# Mesh Interpolation Graph Network for Dynamic and Spatially Irregular Global Weather Forecasting

**Zinan Zheng[1], Yang Liu[2]\*, Jia Li[1]\***
[1]The Hong Kong University of Science and Technology (Guangzhou)
[2]The Chinese University of Hong Kong
zzheng078@connect.hkust-gz.edu.cn
yliuweather@gmail.com, jiale@ust.hk

## Abstract

Graph neural networks have shown promising results in weather forecasting, which is critical for human activity such as agriculture planning and extreme weather preparation. However, most studies focus on finite and local areas for training, overlooking the influence of broader areas and limiting their ability to generalize effectively. Thus, in this work, we study global weather forecasting that is irregularly distributed and dynamically varying in practice, requiring the model to generalize to unobserved locations. To address such challenges, we propose a general **M**esh **I**nterpolation **G**raph **N**etwork (MIGN) that models the irregular weather station forecasting, consisting of two key designs: (1) learning spatially irregular data with regular mesh interpolation network to align the data; (2) leveraging parametric spherical harmonics location embedding to further enhance spatial generalization ability. Extensive experiments on an up-to-date observation dataset show that MIGN significantly outperforms existing data-driven models. Besides, we show that MIGN has spatial generalization ability, and is capable of generalizing to previously unseen stations.

## 1 Introduction

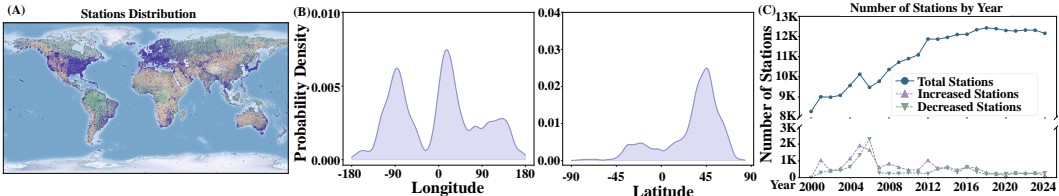

Figure 1: (A). Illustrations of spatially irregular station distribution. (B). The probability density of the station in terms of longitude and latitude. (C). The recorded number of stations in the up-to-date NOAA Global Surface Summary of the Day (GSOD) dataset for each year.

Weather forecasting is critical for human activities and extreme weather warning. For example, accurate short-term predictions of precipitation and snowfall are valuable for agriculture [37] and outdoor activities planning, while forecasting extreme weather phenomena, such as heatwaves [16] and typhoons, is vital to mitigating significant damage. Early warnings can play a crucial role in safeguarding lives and property. To address these problems, multiple date-driven models have been

---

\*Corresponding authors

39th Conference on Neural Information Processing Systems (NeurIPS 2025).

proposed for weather forecasting. A series of works [30, 2, 15, 25] have been developed based on the gridded Earth Reanalysis 5 (ERA5) dataset. However, these models are specifically designed for regular, image-like data structures and cannot be directly applied to weather station data, which consists of precise, fine-grained meteorological observations collected at irregular spatial locations. In contrast, graph neural networks [46, 45, 22, 26, 43, 24, 44, 19, 17, 18, 20, 19] are naturally suited to model such irregular structures. To capture the spatial dependencies inherent in such irregular data, multiple studies have achieved promising results in weather forecasting with GNNs. These approaches typically represent stations as nodes, construct edges among them via radius distance or nearest neighbors, and perform message passing thereon.

However, most of the work [16, 4] focuses on regional forecasting, typically limited to areas such as Europe and North America, while overlooking the influence of external regions. This localized modeling approach overlooks the fact that weather patterns in one region are often influenced by conditions in distant parts of the world, as the Earth's weather system is globally connected. As a result, learning from only regional data often misses broader spatial patterns, leading to suboptimal forecast performance. Moreover, models overfitted to specific regions tend to lack generalization capability, making them less practical for deployment in diverse or unseen geographical areas. Thus, global weather forecasting is crucial and presents the following challenges:

- **Spatial irregularity**. The distribution of weather stations across the Earth's surface is uneven. As illustrated in Figure 1(A), the majority weather stations are concentrated in North America and Western Europe. The spatial distribution of the stations exhibits significant variations in different longitudes and latitudes (shown in Figure 1(B)). Existing data-driven models often overlook the spatial irregularity of station placements, which results in varying scales of information. During training, models often face challenges in simultaneously learning patterns from regions with high and low data point densities.

- **Dynamic distribution**. The number and spatial distribution of stations are changing over time. Figure 1(C) shows the temporal variations in station data of NOAA GSOD dataset[2]. This can be due to the establishment of meteorological stations in remote areas to compensate for limited observational coverage, as well as the decommissioning or abandonment of certain stations over time. Current studies [6, 4, 11, 39] typically use a fixed number of meteorological stations for predictions. Training models on a limited set of stations often results in overfitting of the dataset. Such models often struggle to predict features at unseen locations during training, as the lack of generalization capability limits their performance on previously unobserved points.

To address the above problems, we study a fundamental *spatial generalization* problem in spherical Earth surface. That is, the models are required to predict weather variables in areas with sparse observations or finite historical records. We propose a Mesh Interpolation Graph Network (MIGN) framework that implements a mesh interpolation strategy and parametric spherical harmonics location embedding. To alleviate the uneven distribution of the data, MIGN first maps the latent space of the irregular station to regular mesh by message passing. Such a process could be viewed as interpolation, where the points on the mesh are uniformly distributed. Message passing on mesh points can be implemented to ensure that the model does not only learn patterns from high-density data regions. Secondly, we do not treat the coordinates as position features. Instead, we consider the weather information of the stations as a function of the coordinates, encoding a learnable weather function that can be generalized to unseen points. Through extensive experiments on the up-to-date NOAA GSOD dataset, we find that:

- MIGN outperforms state-of-the-art spatial-temporal models. Ablation studies demonstrate that the two proposed designs, mesh interpolation and spherical harmonic location embedding, significantly enhance the performance.

- The generalization study shows that most methods hard to learn global patterns from existing data, limiting their ability to generalize to unobserved locations. In contrast, MIGN demonstrates strong generalization to unseen stations, highlighting its adaptability to dynamic scenarios.

- Most methods struggle to perform well in regions with dense and sparse observations. In contrast, we show that MIGN consistently produces more robust results across different regional patterns at the same time. The code is available at the link: `https://github.com/compasszzn/MIGN`

---

[2]`https://www.ncei.noaa.gov/access/metadata/landing-page/bin/iso?id=gov.noaa.ncdc:C00516`

## 2 Preliminary and Related Work

**Weather Forecasting**  Traditional weather forecasting depends on Numerical Weather Prediction (NWP) [1] models, which aim to forecast future weather patterns by simulating the dynamics and physics of the atmosphere with the equation of thermodynamics, fluid dynamics, etc. However, NWP requires substantial computing resources and often exhibits deviations [28]. Thus, various data-driven models have been proposed to predict the weather. Currently, data-driven models can be categorized based on the underlying data structure. The first category deals with regular gridded data, with the ECMWF Reanalysis v5 (ERA5) dataset being a representative example. Based on such data, several pioneering works—such as FourCastNet [30], Pangu [2], and GraphCast [15]—have achieved impressive results. However, these models are not well-suited for a second category of data: observed irregular station data. To address this, existing methods often employ Graph Neural Networks (GNNs) to capture spatial dependencies. Nevertheless, these approaches [6, 4, 11, 39] typically assume a fixed set of observation stations over time, limiting their ability to generalize to dynamic scenarios. Motivated by this limitation, we consider a more challenging setting in which the observation stations are irregularly distributed and vary across different samples.

**Problem Definition**  Specifically, we treat each observation station as a node. On day $t$, the global stations could be represented by a graph $\mathcal{G}^t = (\mathcal{V}^t, \mathcal{E}^t, \mathbf{X}^t, (\boldsymbol{\lambda}^t, \boldsymbol{\phi}^t))$, where $\mathcal{V}^t = \left\{ v_1^t, v_2^t, \cdots, v_{|\mathcal{V}^t|}^t \right\}$ is the set of nodes. $\mathcal{E}^t = \{(v_i^t, v_j^t) \mid v_i^t, v_j^t \in \mathcal{V}^t\}$ is edge sets, which is constructed via k-nearest neighbor and the edge attributes (e.g., node distances) are denoted by $d_{ij}$. Each station collects a single weather feature, $\mathbf{X}^t = [x_1^t, x_2^t, \cdots, x_{|\mathcal{V}^t|}^t]$ is a collection of node feature where $x_v^t \in \mathbb{R}, \forall v \in \mathcal{V}^t$. $(\boldsymbol{\lambda}^t, \boldsymbol{\phi}^t)$ denotes global geographic coordinate where longitude $\lambda_i^t \in [-\pi, \pi]$ and latitudes $\phi_i^t \in \left[-\frac{\pi}{2}, \frac{\pi}{2}\right]$. Given the initial condition $\mathcal{G}^t$, our objective is to learn a neural network to predict the next day weather feature value, as shown in the following:

$$\hat{\mathbf{Y}}^{t+1} = f_\Theta(\mathcal{V}^t, \mathcal{E}^t, \mathbf{X}^t, (\boldsymbol{\lambda}^t, \boldsymbol{\phi}^t)), \tag{1}$$

where $\Theta$ denotes the parameters of the neural network. $\hat{\mathbf{Y}}^{t+1}$ denotes the predicted feature while $\mathbf{Y}^{t+1}$ denotes the label. Note that the label $\mathbf{Y}^{t+1}$ here is different to $\mathbf{X}^{t+1}$ because the stations in each step would be different.

**Graph Neural Networks**  Recently, researchers used GNNs to capture spatial patterns of the regional stations, such as air quality estimation[6, 4, 11] and heatwave prediction[16]. The above methods utilize GNNs to capture the spatial correlation and use time series models to model temporal dependency. GNNs are typically implemented using message passing mechanisms. Given a graph $\mathcal{G} = (\mathcal{V}, \mathcal{E}, \mathbf{X}, (\boldsymbol{\lambda}, \boldsymbol{\phi}))$, the message from node $u$ to node $v$ at layer $l$ is given by:

$$\mathbf{m}_{u \to v}^{(l)} = \varphi^{(l)}\left(\mathbf{h}_u^{l-1}, \mathbf{h}_v^{l-1}\right), \forall u \in \mathcal{N}(v), \tag{2}$$

where $\varphi^{(l)}$ can be instantiated as a multi-layer perception (MLP). The generated messages from all neighbors are aggregated at the target node $v$ and the aggregated message $\mathbf{m}_v^{(l)}$ is used to update the state of the node $v$ with function UPDATE$^l$ as follows:

$$\mathbf{m}_v^{(l)} = \text{AGG}^{(l)}\left(\{\mathbf{m}_{u \to v}^{(l)} : u \in \mathcal{N}(v)\}\right), \quad \mathbf{h}_v^l = \text{UPDATE}^l\left(\mathbf{h}_v^{l-1}, \mathbf{m}_v^{(l)}\right), \tag{3}$$

where AGG$^{(l)}$ can be implemented as functions like sum, mean, max pooling or neural network [10, 38, 13, 23] and UPDATE is a learnable function, such as MLP or a gated recurrent unit (GRU).

However, the above framework ignores that the number and spatial distribution of stations change over time. Such a model often fails to predict features in unseen locations.

**Mesh Interpolation**  Mesh interpolation is a common approach in Earth science for using spatially irregular station observation data to reproduce regular mesh data [12, 3]. Traditional interpolation methods include Inverse Distance Weighting (IDW), Kriging, and 3D-thin plate splines (TPS). Among them, IDW is widely used in earth science, which assumes that the influence of a given observation decreases with distance, typically following a power law. Mathematically, the estimated value at an unmeasured location is computed as a weighted average of nearby observations, where the weights are inversely proportional to the distance raised to a specified exponent. Meshes are not only used in

traditional numerical methods but have also been widely adopted in data-driven approaches. One of the most pioneering works in this area is GraphCast [15]. It maps local regions of the input to the nodes of the multi-mesh graph structure and performs message passing on mesh as well. However, it focuses on the regular gridded data and the edges between mesh and nodes are static, while our mesh interpolation lies in alleviating the spatial irregularity problem in station data by mapping the information to a regular space. In addition, the complex distribution of the stations motivates us to enhance the spatial generalization ability of the model. We further propose spherical harmonics location embedding to handle the dynamic data, while GraphCast is based on static data points, which means it lacks generalization capability for grid data with varying resolutions.

**Spherical Harmonics**    The aforementioned GNNs do not incorporate the geometric information of the sphere to improve generalization ability. In contrast, we introduce mesh interpolation to alleviate the spatial irregular problem and spherical harmonics location embedding to enhance spatial generalization. Spherical harmonics have been wides used in earth science for magnetic field [36], weather patterns [40] and gravity field [14]. To be specific, a function $f(\lambda, \phi)$ defined on the sphere can be represented by a set of orthonormalized spherical harmonics $Y_n^m(\lambda, \phi)$ as follows:

$$f(\lambda, \phi) = \sum_{n=0}^{\infty} \sum_{m=-n}^{n} w_n^m Y_n^m(\lambda, \phi), \tag{4}$$

where $n$ denotes the degree, which controls the spatial scale of variation, with small $n$ capturing coarse, global patterns and larger $n$ resolving finer structures. $m$ denotes the order with $m \in [-n, n]$ of the basis functions, governing the oscillations in the longitudinal direction. $\lambda$ and $\phi$ are longitude and latitude respectively. We consider a maximum degree of $N$, which results in a total of $(N + 1)^2$ basis functions and learnable weights $w_n^m$. The spherical harmonics are functions defined on the sphere as:

$$Y_n^m(\lambda, \phi) = \sqrt{\frac{2n+1}{4\pi} \frac{(n-|m|)!}{(n+|m|)!}} P_n^m(\cos \lambda) e^{im\phi}, \tag{5}$$

where $P_n^m$ are associated Legendre polynomials:

$$P_n^m(x) = (-1)^m (1 - x^2)^{|m|/2} \frac{d^{|m|}}{dx^{|m|}} P_n(x), \tag{6}$$

which involve derivatives of Legendre Polynomials $P_n(x)$ defined by the following recurrence:

$$P_0(x) = 1, P_1(x) = x, nP_n(x) = (2n-1)xP_{n-1}(x) - (n-1)P_{n-2}(x). \tag{7}$$

In practice, we consider the real spherical harmonics given as

$$Y_n^m(\lambda, \phi) = \hat{P}_n^{|m|}(\cos \lambda) \cdot \begin{cases} \sin(|m|\phi) & m < 0 \\ 1 & m = 0 \\ \cos(m\phi) & m > 0. \end{cases} \tag{8}$$

where $\hat{P}_n^{|m|}(\cos \lambda) = \sqrt{\frac{2n+1}{4\pi} \frac{(n-m)!}{(n+m)!}} P_n^{|m|}(\cos \lambda)$, following the work [32], we pre-compute the spherical harmonics for each node in experiments. A related work is Geographic Location Encoder [32]. Although Geographic Location Encoder utilizes spherical harmonics, it focuses on training a neural network based on land-ocean classification tasks for coordinate embedding and spatial forecasting (i.e., ERA5 interpolation) of weather data to learn the coefficients of the spherical harmonics. However, MIGN aims to spatio-temporal forecast of irregular and dynamic distributed weather station data, therefore, it employs the spherical harmonic embedding as part of the input. Besides, considering that the variation patterns of different weather variables differ within the same region, we would learn a different variable-specific location embedding.

## 3    Method

Our MIGN architecture is illustrated in Figure 2, following an encoder-processor-decoder framework. In the following, we elaborate on the MIGN framework including spherical harmonics location embedding and mesh interpolation.

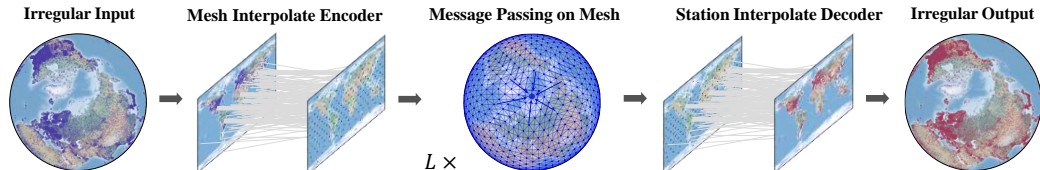

Figure 2: Framework of the model. MIGN architecture follows an encoder-processor-decoder framework.

### 3.1 Spherical Harmonics Location Embedding

Spherical harmonics are widely used in the analysis of global weather patterns [40]. Since the Earth can be approximated as a sphere, many meteorological variables can be naturally modeled as functions defined on the spherical surface. Spherical harmonics provide a convenient basis for representing such functions, allowing us to capture spatial structures of the station. Besides, the non-parametric positional embedding provides limited location information, which restricts the model's ability to generalize to unseen areas. Inspired by this, we assume that the global location information could be represented by a function $f(\lambda, \phi)$ defined on the sphere. Instead of learning this function with a neural network directly, we decomposed this function into spherical harmonics to learn the spherical harmonics coefficients. The figure for the method is shown in the Appendix Figure 7. Using real spherical harmonics, the function of the sphere could be represented by $f(\lambda, \phi) = \sum_{n=0}^{\infty} \sum_{m=-n}^{n} w_n^m Y_n^m(\lambda, \phi)$, $w_n^m$ refers to spherical harmonics coefficients, which are learnable weights.

Consider the expressive power of the location embedding. We concatenate the spherical harmonic basis function as features instead of learning the function $f(\lambda, \phi)$ directly. Specifically, consider the node with feature $x$ and coordinates $\lambda, \phi$. The location embedding is denoted as follows:

$$SH(\lambda, \phi) = \bigoplus_{n \geq 0} \Big( \bigoplus_{n \geq m \geq -n} \Big( w_n^m Y_n^m(\lambda, \phi) \Big) \Big), \quad \mathbf{h} = [x; SH(\lambda, \phi)], \tag{9}$$

where $\bigoplus$ indicates the concatenation of these basis functions into one large vector and $[;]$ denotes the concatenation of the embedding vector. Based on the definition of spherical harmonics, the learnable $w_n^m$ coefficient is shared across all nodes.

### 3.2 MIGN Framework

In this section, we first introduce the HEALPix which is employed to construct mesh. Then we would elaborate on the mesh interpolation framework with spherical harmonics location embedding.

**HEALPix Mesh**  HEALPix [8] (Hierarchical, Equal Area, and iso-Latitude Pixelisation of the sphere) is a hierarchical structure for multi-resolution applications, which uniformly divides the sphere into equal-sized pixels. The data points are located in the center of the pixels and are uniformly distributed across the sphere. The base resolution consists of 12 quadrilateral pixels on the sphere. To generate a higher-resolution HEALPix grid, each pixel can be subdivided along the edge twice, resulting in 4 subgrids that represent the original quadrilateral pixel. We can do this recursively to get a higher-resolution HEALPix mesh. When the process is conducted $k$ times, which is also called refinement level $k$, the original quadrilateral pixel can be divided into $(2^k)^2$ pixels, leading to $12 * (2^k)^2$ pixels and mesh nodes in total.

**Mesh Interpolate Encoder**  Irregular station distributions make it hard to represent spatial patterns, and graph-based neighborhood aggregation becomes difficult due to the lack of consistent locality and connectivity. Motivated by this, MIGN first conducts a message passing from station nodes to regular mesh nodes within the encoder. Consider the graph in the encoder at day t $\mathcal{G}_E^t = ((\mathcal{V}_s^t, \mathcal{V}_h^t), \mathcal{E}_{(s,h)}^t, (\mathbf{X}_s^t, \mathbf{X}_h^t), (\boldsymbol{\lambda}_s^t, \boldsymbol{\phi}_s^t))$, where label $s$ denotes station nodes while label $h$ denotes the mesh nodes. The feature of the mesh nodes $\mathbf{X}_h$ can be initialized with zero. $\mathcal{E}_{(s,h)}^t = \{(v_s^t, v_h^t) \mid$

$v_s^t \in \mathcal{V}_s^t, v_h^t \in \mathcal{V}_h^t\}$ is edge sets. Inspired by mesh interpolation in earth science, we only consider constructing the edges from station nodes to mesh nodes. Instead of interpolating the value of mesh nodes with fixed weight like IDW, we utilized message passing neural network to project the value into latent space. For each mesh node $v_h^t$, messages are generated by its neighbors station nodes $v_s^t \in \mathcal{N}(v_h^t)$. The hidden state of the station nodes and the message are given by:

$$\mathbf{h}_{v_s^t} = [x_{v_s^t}^t; SH(\lambda_{v_s^t}^t, \phi_{v_s^t}^t)], \quad \mathbf{m}_{v_s^t \to v_h^t}^{(E)} = \varphi^{(E)}\left(\mathbf{h}_{v_s^t}\right), \forall v_s^t \in \mathcal{N}(v_h^t), \tag{10}$$

**Message Passing** The messages from the station nodes are aggregated to the target mesh nodes, and the hidden state of the mesh nodes would update with the message directly:

$$\mathbf{h}_{v_h^t}^{(E)} = \text{AGG}^{(E)}\left(\{\mathbf{m}_{v_s^t \to v_h^t}^{(E)} : v_s^t \in \mathcal{N}(v_h^t)\}\right), \tag{11}$$

For the processor part, we consider the mesh nodes graph $\mathcal{G}_P^t = (\mathcal{V}_h^t, \mathcal{E}_h^t, \mathbf{H}_h^t, (\boldsymbol{\lambda_h^t}, \boldsymbol{\phi_h^t}))$. The feature of the mesh nodes $\mathbf{H}_h^t$ are the message aggregate from station nodes, denoted as $\mathbf{h}_{v_h^t}^{(E)}, v_h^t \in \mathcal{V}_h^t$ for each mesh node. The hidden state of the $0th$ layer processor and the message are denoted as

$$\mathbf{h}_{v_h^t}^{(0)} = [\mathbf{h}_{v_h^t}^{(E)}; SH(\lambda_{v_h^t}^t, \phi_{v_h^t}^t)], \quad \mathbf{m}_{v_{\hat{h}}^t \to v_h^t}^{(l)} = \varphi^{(l)}\left(\mathbf{h}_{v_{\hat{h}}^t}^{l-1}, \mathbf{h}_{v_h^t}^{l-1}\right), \forall v_{\hat{h}}^t \in \mathcal{N}(v_h^t), \tag{12}$$

Messages are exchanged within the mesh nodes and aggregated as follows:

$$\mathbf{m}_{v_h^t}^{(l)} = \text{AGG}^{(l)}\left(\{\mathbf{m}_{v_{\hat{h}}^t \to v_h^t}^{(l)} : v_{\hat{h}}^t \in \mathcal{N}(v_h^t)\}\right), \quad \mathbf{h}_{v_h^t}^l = \text{UPDATE}^l\left(\mathbf{h}_{v_h^t}^{l-1}, \mathbf{m}_{v_h^t}^{(l)}\right), \tag{13}$$

The output hidden state of the processor is denoted as $\mathbf{H}_h^{t+1}$, which refers to the latent space of the mesh in the next time step. On the regular mesh, spatial adjacency is clearly defined, and each node has a fixed position. This allows for standard modeling tools (e.g., CNNs, GNNs, Transformers) to be used effectively. Any existing GNN can be implemented in these phases, which makes MIGN a flexible method. Because the spatial layout of the mesh remains fixed over time, it provides a consistent data structure across time steps, enabling more stable and coherent temporal modeling.

**Station Interpolate Decoder** After modeling on the mesh, the results need to be mapped back to the observation stations to enable comparison with real-world measurements. The decoder follows a reverse process of the encoder. Consider the graph in the decoder $\mathcal{G}_D^{t+1} = \left((\mathcal{V}_h^{t+1}, \mathcal{V}_s^{t+1}), \mathcal{E}_{(h,s)}, \left(\mathbf{H}_h^{t+1}, \hat{\boldsymbol{Y}}_s^{t+1}\right), (\lambda_h^t, \phi_h^t)\right)$, $\hat{\boldsymbol{Y}}_s^{t+1}$ denotes the predicted feature in next step. The decoder would aggregate the message from the hidden state of a $L$ layer processor directly to update the $\hat{\boldsymbol{Y}}_s^{t+1}$ as follows:

$$\mathbf{h}_{v_h^{t+1}} = [\mathbf{h}_{v_h^t}^L; \lambda_{v_h^t}^t; \phi_{v_h^t}^t], \quad \mathbf{m}_{v_h^{t+1} \to v_s^{t+1}}^{(D)} = \varphi^{(D)}\left(\mathbf{h}_{v_h^{t+1}}\right), \forall v_h^{t+1} \in \mathcal{N}(v_s^{t+1}),$$
$$\hat{\mathbf{y}}_{v_s^{t+1}} = \text{AGG}^{(D)}\left(\{\mathbf{m}_{v_h^{t+1} \to v_s^{t+1}}^{(D)} : v_h^{t+1} \in \mathcal{N}(v_s^{t+1})\}\right). \tag{14}$$

Our framework is readily adaptable to multi-step input and output, as shown in Appendix A.4.

**Training** Given the predicted feature $\hat{\boldsymbol{Y}}_s^{t+1}$ of the decoder. The model parameters can be optimized by minimizing the discrepancy between the prediction and ground truth: $\mathcal{L}_{\text{train}} = \sum_{s \in \mathcal{D}_{\text{train}}} ||\hat{\boldsymbol{Y}}_s^{t+1} - \boldsymbol{Y}_s^{t+1}||^2$.

### 3.3 Generalization Empirical Verification

To illustrate our motivation, we conduct global generalization experiments. Specifically, we randomly sample half of the stations from 2017–2023 for training and validation, while reserving the unseen half from 2024 as the test set. Detailed experimental settings are provided in Section 4.3. The results for mean sea level pressure (SLP) are visualized in Figure 3. As shown, predictions from both DyGrAE and STAR exhibit higher MAE values across large regions of Europe and North America, indicating that these baseline models struggle to generalize to previously unobserved areas. In contrast, MIGN achieves lower errors in these regions, demonstrating superior generalization performance. A complete numerical comparison is provided in Table 4.

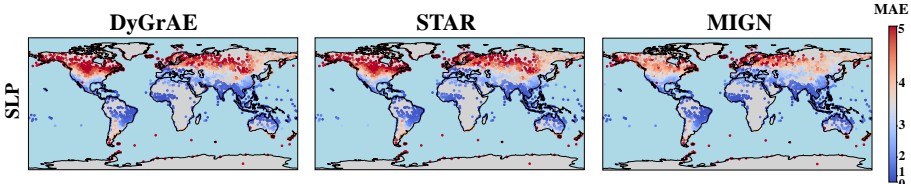

Figure 3: The global MAE distribution of SLP in the generalization experiment testing set

# 4 Experiments

**Dataset** We evaluate model performance on a up-to-date daily NOAA Global Surface Summary of the Day (GSOD) dataset. We use 6 commonly daily observed variables, including maximum temperature (MAX TEMP), minimum temperature (MIN TEMP), mean dew point (DEWP), mean sea level pressure (SLP), mean wind speed (WDSP) and maximum sustained wind speed (MXSPD). We use the 2017-2022 data for training, the 2023 data for validation, and the 2024 for testing. The detailed information of the dataset is shown in the Appendix A.6.

**Baselines and metric** We compare our MIGN with the following 13 spatial-temporal baselines: (1) global-based models: STGCN [41], MPNNLSTM [29], DualCast [34]. (2) global-local based models: T&S-IMP, T&S-AMP, TTS-IMP, TTS-AMP [5]. (3) dynamic graph models: DyGrAE [35], ReDyNet [7]. (4) graph pooling based models: HD-TTS [27]. (5) Transformer based models: STAR [42], GTN [33], GPS [31]. We adopt Mean Squared Error (MSE) and (Mean Absolute Error) MAE to evaluate the model performance. We run each method five times and report the average metric of all models.

**Implementation** We utilize Adam optimizer to train our model and use the following hyperparameters: Batch size 4, hidden state 64, and learning rate 0.001. The model is set to 2 layers. The mesh refinement level is set to 3, and we use 10-nearest neighbor to construct the graph, and the spherical harmonics degree is set to 2. All models are implemented based on Pytorch Lightning, trained on GeForce RTX3090 GPU. Baseline models are implemented with PyG library, while our model is realized with the DGL library. For a fair comparison, we tune different hyperparameters for all baselines, finding the best setting for each. The detailed information can be found in the Appendix A.7 and Tabel7.

## 4.1 Overall Performance

In this section, we evaluate the performance of our proposed model against several baseline methods. As summarized in Table 1, our approach consistently outperforms all baselines across every variable. In particular, MIGN achieves relative MSE improvements of 13%, 15%, and 15% on MAX TEMP, MIN TEMP, and SLP, respectively, compared to the strongest baseline. To further evaluate our model's performance across different time horizons, we conduct experiments using a three-day multistep input and a four-day multistep output training setup. The results, summarized in Table 2, show that our proposed MIGN consistently outperforms all baselines across both short- and long-term horizons, highlighting its robustness and effectiveness in the multistep forecasting setting. Besides, we further conduct a series of studies, including varying input steps, autoregressive inference. The results are presented in Appendix A.8.1.

## 4.2 Ablation Study

To demonstrate the effectiveness of each model design, we compare the default configuration of MIGN with four variants that differ in their use of spherical harmonics location embedding and mesh interpolation. As shown in Table 3, we observe that: (1) adopting mesh interpolation consistently improves performance; for example, DEWP and SLP MSE decrease from 9.00/23.93 to 7.92/20.09. (2) spherical harmonics embedding further enhances performance when applied to both the encoder and decoder, as the encoder embedding captures station node locations while the decoder embedding represents mesh node locations. This validates the effectiveness of spherical harmonics embeddings

Table 1: Bold font indicates the best result, and Underline is the strongest baseline. We report both the mean and the standard deviation that are computed over 5 runs.

| Model | MAX TEMP ($K$) MSE | MAE | MIN TEMP ($K$) MSE | MAE | DEWP ($K$) MSE | MAE | SLP ($mb$) MSE | MAE | WDSP ($kn$) MSE | MAE | MXSPD ($kn$) MSE | MAE |
|---|---|---|---|---|---|---|---|---|---|---|---|---|
| Persistence | 9.98 | 2.17 | 9.80 | 2.09 | 9.56 | 2.10 | 26.62 | 3.54 | 10.35 | 2.15 | 25.32 | 3.48 |
| STGCN (2017) | 9.74±0.00 | 2.22±0.00 | 9.44±0.00 | 2.11±0.00 | 9.25±0.00 | 2.11±0.00 | 24.15±0.00 | 3.42±0.00 | 8.60±0.00 | 2.01±0.00 | 20.63±0.00 | 3.27±0.00 |
| DyGrAE (2019) | 10.13±0.33 | 2.24±0.02 | 9.49±0.08 | 2.11±0.02 | 9.25±0.05 | 2.10±0.01 | 24.09±0.05 | 3.40±0.00 | 8.77±0.03 | 2.04±0.01 | 20.78±0.07 | 3.28±0.01 |
| STAR (2020) | 10.18±0.00 | 2.26±0.00 | 9.65±0.01 | 2.15±0.00 | 9.56±0.01 | 2.16±0.00 | 24.14±0.00 | 3.42±0.00 | 9.31±0.00 | 2.10±0.00 | 21.88±0.00 | 3.36±0.00 |
| GTN (2020) | 9.88±0.00 | 2.22±0.00 | 9.49±0.00 | 2.14±0.00 | 9.51±0.00 | 2.16±0.00 | 24.49±0.00 | 3.44±0.00 | 8.82±0.00 | 2.01±0.00 | 20.86±0.08 | 3.30±0.03 |
| MPNNLSTM (2021) | 47.34±0.13 | 4.70±0.06 | 45.24±0.01 | 4.46±0.00 | 40.94±0.08 | 4.33±0.00 | 38.74±0.03 | 4.40±0.02 | 10.48±0.00 | 2.33±0.00 | 24.66±0.01 | 3.69±0.00 |
| GPS (2022) | 10.91±0.49 | 2.42±0.08 | 10.37±0.39 | 2.27±0.05 | 11.13±0.32 | 2.50±0.06 | 25.24±1.14 | 3.57±0.16 | 8.79±0.11 | 2.04±0.01 | 20.89±0.06 | 3.30±0.01 |
| T&S-IMP (2023) | 12.12±0.70 | 2.51±0.08 | 10.92±0.58 | 2.33±0.08 | 10.80±0.10 | 2.31±0.02 | 24.70±0.21 | 3.46±0.02 | 8.88±0.06 | 2.06±0.02 | 20.93±0.16 | 3.30±0.01 |
| T&S-AMP (2023) | 10.16±0.10 | 2.28±0.03 | 12.90±2.65 | 2.59±0.33 | 9.43±0.10 | 2.16±0.03 | 24.38±0.16 | 3.44±0.02 | 8.88±0.10 | 2.04±0.02 | 20.72±0.25 | 3.28±0.02 |
| TTS-IMP (2023) | 10.40±0.10 | 2.32±0.03 | 11.58±0.96 | 2.44±0.11 | 13.69±3.66 | 2.64±0.35 | 24.76±0.28 | 3.47±0.03 | 9.05±0.16 | 2.07±0.02 | 21.86±0.60 | 3.40±0.05 |
| TTS-AMP (2023) | 9.88±0.22 | 2.25±0.02 | 9.80±0.06 | 2.18±0.02 | 9.91±0.00 | 2.19±0.00 | 24.43±0.13 | 3.45±0.02 | 8.74±0.08 | 2.05±0.02 | 20.79±0.35 | 3.28±0.03 |
| HD-TTS (2024) | 10.20±0.01 | 2.33±0.00 | 9.65±0.02 | 2.17±0.01 | 9.77±0.02 | 2.21±0.01 | 24.27±0.10 | 3.44±0.02 | 9.11±0.29 | 2.11±0.05 | 20.25±0.21 | 3.23±0.01 |
| ReDyNet (2025) | 10.33±0.05 | 2.26±0.01 | 10.85±0.15 | 2.30±0.04 | 10.81±0.12 | 2.32±0.04 | 24.15±0.11 | 3.40±0.02 | 8.75±0.05 | 2.06±0.01 | 20.95±0.17 | 3.28±0.04 |
| DualCast (2025) | 10.84±0.08 | 2.40±0.02 | 10.11±0.09 | 2.26±0.03 | 9.42±0.08 | 2.15 ±0.02 | 23.83±0.04 | 3.39±0.00 | 8.63±0.13 | 2.03±0.02 | 20.27±0.15 | 3.25±0.01 |
| MIGN | **8.47±0.05** | **2.09±0.01** | **8.01±0.04** | **1.99±0.01** | **7.92±0.05** | **1.97±0.01** | **20.09±0.07** | **3.12±0.01** | **8.38±0.01** | **1.98±0.01** | **19.73±0.05** | **3.19±0.01** |
| Improvements | 13% | 4% | 15% | 5% | 15% | 6% | 15% | 8% | 3% | 2% | 3% | 2% |

Table 2: Bold font indicates the best result, and Underline is the strongest baseline. We report the mean MSE that is computed over 5 runs.

| Model | MAX TEMP($K$) Step1 | Step2 | Step3 | Step4 | Total | MIN TEMP ($K$) Step1 | Step2 | Step3 | Step4 | Total | DEWP($K$) Step1 | Step2 | Step3 | Step4 | Total | SLP ($mb$) Step1 | Step2 | Step3 | Step4 | Total |
|---|---|---|---|---|---|---|---|---|---|---|---|---|---|---|---|---|---|---|---|---|
| Persistence | 9.98 | 18.58 | 23.43 | 26.36 | 19.60 | 9.80 | 17.94 | 22.21 | 24.41 | 18.63 | 9.56 | 19.82 | 24.87 | 27.47 | 20.49 | 26.62 | 58.70 | 74.68 | 84.31 | 61.16 |
| STGCN (2017) | 11.10 | 16.87 | 20.09 | 22.05 | 17.53 | 10.38 | 15.65 | 18.36 | 19.96 | 16.09 | 10.42 | 17.15 | 20.15 | 22.23 | 17.49 | 22.25 | 42.93 | 51.08 | 55.22 | 42.87 |
| DyGrAE (2019) | 9.85 | 16.81 | 20.40 | 22.49 | 17.39 | 9.27 | 15.29 | 18.19 | 19.72 | 15.58 | 9.01 | 16.91 | 20.22 | 21.87 | 17.00 | 23.93 | 44.18 | 52.30 | 56.58 | 44.25 |
| STAR (2020) | 9.92 | 16.73 | 20.43 | 24.45 | 17.88 | 9.75 | 15.81 | 18.46 | 20.16 | 16.05 | 11.06 | 17.94 | 21.82 | 22.96 | 18.45 | 22.86 | 44.85 | 53.79 | 58.20 | 44.93 |
| GTN (2020) | 10.43 | 16.94 | 20.76 | 22.86 | 17.75 | 9.94 | 16.25 | 22.27 | 22.65 | 17.78 | 9.74 | 18.01 | 21.07 | 23.35 | 18.04 | 23.09 | 47.23 | 53.78 | 57.21 | 45.33 |
| MPNNLSTM (2021) | 45.49 | 50.04 | 52.18 | 53.20 | 50.23 | 45.29 | 47.81 | 49.10 | 49.59 | 47.95 | 40.55 | 44.90 | 46.62 | 46.86 | 44.68 | 37.58 | 54.59 | 61.73 | 65.89 | 54.95 |
| GPS (2022) | 12.29 | 18.43 | 21.86 | 24.03 | 19.15 | 10.65 | 16.37 | 19.26 | 20.80 | 16.77 | 11.46 | 18.21 | 21.66 | 23.15 | 18.62 | 22.45 | 43.79 | 51.88 | 56.05 | 43.54 |
| T&S-IMP (2023) | 12.28 | 18.17 | 21.44 | 23.57 | 18.86 | 11.37 | 16.67 | 19.39 | 20.77 | 17.05 | 9.97 | 17.57 | 20.79 | 22.39 | 17.68 | 23.54 | 43.83 | 51.77 | 56.00 | 43.78 |
| T&S-AMP (2023) | 10.65 | 16.77 | 20.17 | 22.18 | 17.44 | 10.28 | 15.60 | 18.29 | 19.78 | 15.99 | 11.12 | 18.11 | 21.70 | 23.63 | 18.64 | 22.62 | 42.87 | 51.06 | 55.49 | 43.01 |
| TTS-IMP (2023) | 11.69 | 17.72 | 20.92 | 22.84 | 18.29 | 10.51 | 15.79 | 18.59 | 20.04 | 16.23 | 12.22 | 19.04 | 22.45 | 24.15 | 19.47 | 24.17 | 44.52 | 52.95 | 56.90 | 44.64 |
| TTS-AMP (2023) | 9.59 | 16.37 | 19.70 | 21.61 | 16.82 | 10.82 | 16.29 | 19.04 | 20.62 | 16.69 | 10.28 | 17.13 | 20.40 | 22.29 | 17.53 | 22.94 | 43.07 | 51.44 | 56.07 | 43.38 |
| HD-TTS (2024) | 10.07 | 16.47 | 19.78 | 21.71 | 17.00 | 10.65 | 16.00 | 18.72 | 20.17 | 16.39 | 10.71 | 17.89 | 21.48 | 23.27 | 18.34 | 22.84 | 44.00 | 52.09 | 56.12 | 43.76 |
| ReDyNet (2025) | 17.89 | 21.72 | 23.90 | 25.62 | 22.28 | 16.72 | 20.14 | 21.63 | 22.96 | 20.36 | 18.71 | 22.61 | 24.62 | 25.54 | 22.87 | 47.97 | 56.31 | 60.42 | 63.10 | 56.95 |
| DualCast (2025) | 10.05 | 16.50 | 19.87 | 21.89 | 17.08 | 10.24 | 15.78 | 18.44 | 19.78 | 16.06 | 10.14 | 17.57 | 20.64 | 22.16 | 17.63 | 22.41 | 43.37 | 51.21 | 54.90 | 42.97 |
| MIGN | **8.41** | **14.62** | **18.27** | **20.58** | **15.47** | **9.20** | **14.88** | **17.68** | **19.50** | **15.33** | **8.19** | **15.47** | **19.02** | **21.23** | **15.98** | **19.29** | **39.93** | **48.99** | **53.37** | **40.40** |

in learning geometric geographic information from data. For completeness, we also compare our SH embedding with the commonly used coordinate-based embedding, with results reported in Appendix A.8.2.

## 4.3 Global Generalization Analysis

To evaluate model performance in a global and dynamic setting, we further conduct an experiment to validate the generalization ability of MIGN. We randomly sample half of the stations from the year 2017-2022/2023 for training and validation, while using the remaining stations from 2024 as the test set. Although the global distribution of stations is similar between the training and test sets, the test stations are entirely unseen during training. As shown in Table 4, We can find that MIGN outperforms all baselines across all variables, achieving the lowest MSE and MAE consistently. For example, MIGN achieves an MSE of 8.55/8.05 in MAX TEMP and MIN TEMP, outperforming the closest baseline 9.81/9.52 respectively. These results highlight MIGN's superior ability to generalize to unobserved stations in dynamic, real-world scenarios.

## 4.4 Sparse region analysis

To investigate the model performance in area with sparse weather station coverage, we analyze the model performance in data-scarce regions, including Africa, Asia, Australia, and South America, as shown in Figure 4. Across all regions and variables, MIGN consistently achieves the lowest MSE, highlighting its strong generalization capability in low-resource environments. Notably, in Asia, MIGN demonstrates significant improvements, reducing the MSE for MAX TEMP and MIN TEMP to below 8 and 6, respectively—thresholds that other models fail to surpass. These findings suggest that MIGN effectively captures variable patterns even under sparse observational conditions.

## 4.5 Mesh Analysis

**Refinement level analysis** To validate the effect of different refinement level mesh on the MIGN performance. We compare the metric of 5 different refinement levels (corresponding 48, 192, 768,

Table 3: Ablation studies.

| Model Variant | MAX TEMP (K) | | MIN TEMP (K) | | DEWP (K) | | SLP (mb) | | WDSP (kn) | | MXSPD (kn) | |
|---|---|---|---|---|---|---|---|---|---|---|---|---|
| | MSE | MAE | MSE | MAE | MSE | MAE | MSE | MAE | MSE | MAE | MSE | MAE |
| w/o mesh & SH (MPNN) | 9.82±0.01 | 2.23±0.01 | 9.78±0.04 | 2.16±0.01 | 9.40±0.053 | 2.13±0.01 | 24.27±0.04 | 3.42±0.02 | 8.85±0.08 | 2.04±0.01 | 21.12±0.01 | 3.34±0.01 |
| w/o mesh (MPNN+SH) | 9.48±0.02 | 2.19±0.00 | 9.04±0.02 | 2.08±0.01 | 9.00±0.05 | 2.08±0.01 | 23.93±0.05 | 3.39±0.01 | 8.74±0.01 | 2.04±0.01 | 20.77±0.02 | 3.26±0.01 |
| w/o SH | 9.04±0.08 | 2.15±0.01 | 8.71±0.05 | 2.06±0.01 | 8.71±0.04 | 2.06±0.01 | 23.01±0.07 | 3.33±0.01 | 8.76±0.02 | 2.03±0.01 | 20.63±0.04 | 3.27±0.01 |
| w/o encoder SH | 8.80±0.07 | 2.12±0.01 | 8.52±0.06 | 2.04±0.01 | 8.56±0.07 | 2.04±0.01 | 22.57±0.11 | 3.29±0.01 | 8.59±0.04 | 2.01±0.01 | 20.19±0.03 | 3.23±0.01 |
| w/o decoder SH | 8.60±0.05 | 2.12±0.01 | 8.20±0.04 | 2.01±0.01 | 7.99±0.03 | 1.98±0.01 | 22.07±0.09 | 3.21±0.01 | 8.39±0.04 | 1.98±0.01 | 19.78±0.05 | 3.22±0.01 |
| Default | **8.47±0.05** | **2.09±0.01** | **8.01±0.04** | **1.99±0.01** | **7.92±0.05** | **1.97±0.01** | **20.09±0.07** | **3.12±0.01** | **8.38±0.01** | **1.98±0.01** | **19.73±0.05** | **3.19±0.01** |
| Improvements | 14% | 6% | 18% | 8% | 16% | 8% | 17% | 9% | 5% | 3% | 7% | 4% |

Table 4: Bold font indicates the best result and Underline is the strongest baseline. We report the mean results that are computed over 5 runs. Global generalization experiments.

| Model | MAX TEMP (K) | | MIN TEMP (K) | | DEWP (K) | | SLP (mb) | | WDSP (kn) | | MXSPD (kn) | |
|---|---|---|---|---|---|---|---|---|---|---|---|---|
| | MSE | MAE | MSE | MAE | MSE | MAE | MSE | MAE | MSE | MAE | MSE | MAE |
| Persistence | 9.98 | 2.17 | 9.78 | 2.09 | 9.65 | 2.11 | 26.69 | 3.54 | 10.31 | 2.15 | 25.45 | 3.48 |
| STGCN (2017) | 9.87±0.00 | 2.23±0.00 | 9.52±0.00 | 2.15±0.00 | 9.45±0.00 | 2.14±0.00 | 25.81±0.00 | 3.59±0.00 | 8.62±0.00 | 2.02±0.00 | 20.90±0.00 | 3.32±0.00 |
| DyGrAE (2019) | 10.83±0.22 | 2.27±0.02 | 9.55±0.02 | 2.12±0.00 | 9.53±0.02 | 2.13±0.00 | 24.40±0.46 | 3.41±0.03 | 8.78±0.03 | 2.03±0.01 | 21.01±0.01 | 3.28±0.00 |
| STAR (2020) | 9.99±0.00 | 2.24±0.00 | 9.55±0.00 | 2.15±0.00 | 9.54±0.00 | 2.14±0.00 | 24.25±0.00 | 3.42±0.00 | 8.99±0.00 | 2.06±0.00 | 21.67±0.00 | 3.36±0.00 |
| GTN (2020) | 9.89±0.00 | 2.23±0.00 | 9.56±0.00 | 2.15±0.00 | 9.66±0.00 | 2.18±0.00 | 24.55±0.00 | 3.44±0.00 | 8.84±0.00 | 2.00±0.00 | 21.01±0.09 | 3.30±0.03 |
| MPNNLSTM (2021) | 51.15±0.25 | 5.07±0.09 | 49.09±0.03 | 4.71±0.01 | 44.61±0.03 | 4.65±0.00 | 41.00±0.01 | 4.50±0.00 | 10.66±0.00 | 2.41±0.00 | 25.15±0.01 | 3.73±0.00 |
| GPS (2022) | 13.90±3.67 | 2.79±0.45 | 11.50±1.52 | 2.45±0.19 | 10.54±0.61 | 2.36±0.15 | 24.97±1.12 | 3.52±0.14 | 8.82±0.17 | 2.06±0.05 | 21.03±0.12 | 3.30±0.04 |
| T&S-IMP (2023) | 12.11±0.75 | 2.46±0.06 | 12.11±0.85 | 2.45±0.10 | 11.45±0.22 | 2.34±0.02 | 24.99±0.34 | 3.48±0.03 | 8.93±0.06 | 2.08±0.00 | 21.29±0.07 | 3.33±0.00 |
| T&S-AMP (2023) | 10.38±0.17 | 2.30±0.03 | 10.97±0.30 | 2.35±0.03 | 9.78±0.08 | 2.17±0.02 | 24.70±0.17 | 3.47±0.02 | 8.85±0.00 | 2.05±0.01 | 20.88±0.04 | 3.29±0.01 |
| TTS-IMP (2023) | 10.53±0.23 | 2.33±0.03 | 10.42±0.61 | 2.27±0.09 | 16.22±3.83 | 2.38±0.12 | 24.88±0.38 | 3.47±0.03 | 8.96±0.04 | 2.07±0.01 | 21.66±0.68 | 3.32±0.02 |
| TTS-AMP (2023) | 11.30±1.56 | 2.43±0.21 | 9.80±0.05 | 2.17±0.02 | 10.15±0.00 | 2.23±0.00 | 24.61±0.15 | 3.45±0.02 | 8.84±0.12 | 2.06±0.02 | 21.31±0.22 | 3.32±0.01 |
| HD-TTS (2024) | 9.81±0.19 | 2.25±0.04 | 9.71±0.04 | 2.18±0.03 | 9.58±0.07 | 2.14±0.02 | 24.39±0.06 | 3.44±0.01 | 8.96±0.01 | 2.09±0.01 | 21.55±0.10 | 3.36±0.01 |
| ReDyNet (2025) | 10.41±0.04 | 2.31±0.01 | 10.97±0.06 | 2.38±0.02 | 10.97±0.18 | 2.51±0.02 | 24.31±0.15 | 3.52±0.02 | 8.92±0.04 | 2.13±0.01 | 21.09±0.09 | 3.32±0.04 |
| DualCast (2025) | 10.91±0.02 | 2.43±0.02 | 10.38±0.07 | 2.33±0.01 | 9.49±0.06 | 2.17±0.04 | 23.87±0.02 | 3.42±0.00 | 8.68±0.05 | 2.08±0.01 | 20.32±0.11 | 3.29±0.01 |
| MIGN | **8.55±0.10** | **2.10±0.01** | **8.05±0.14** | **2.00±0.02** | **7.95±0.08** | **1.99±0.01** | **20.90±0.13** | **3.14±0.02** | **8.34±0.04** | **1.98±0.01** | **19.82±0.07** | **3.20±0.01** |

Table 5: Spherical Harmonics degree analysis.

| Degree | Order | MAX TEMP (K) | | MIN TEMP (K) | | DEWP (K) | | SLP (mb) | | WDSP (kn) | | MXSPD (kn) | |
|---|---|---|---|---|---|---|---|---|---|---|---|---|---|
| | | MSE | MAE | MSE | MAE | MSE | MAE | MSE | MAE | MSE | MAE | MSE | MAE |
| 0 | 1 | 8.99±0.05 | 2.14±0.01 | 8.71±0.14 | 2.06±0.00 | 8.71±0.02 | 2.06±0.01 | 23.01±0.06 | 3.33±0.01 | 8.76±0.02 | 2.03±0.00 | 20.56±0.04 | 3.26±0.01 |
| 1 | 4 | 8.82±0.04 | 2.13±0.01 | 8.44±0.12 | 2.02±0.01 | 8.29±0.04 | 2.01±0.01 | 21.75±0.05 | 3.24±0.01 | 8.44±0.02 | 1.99±0.01 | 19.92±0.03 | 3.20±0.01 |
| 2 | 9 | 8.47±0.05 | **2.09±0.01** | 8.01±0.04 | **1.99±0.01** | 7.92±0.05 | 1.97±0.01 | 20.09±0.07 | 3.12±0.01 | 8.38±0.01 | **1.98±0.01** | 19.73±0.05 | **3.19±0.01** |
| 3 | 16 | **8.38±0.10** | **2.09±0.01** | 8.16±0.08 | 2.01±0.01 | **7.76±0.04** | **1.95±0.01** | **20.06±0.05** | **3.11±0.02** | 8.35±0.03 | 1.99±0.01 | **19.67±0.05** | **3.19±0.01** |

3072 and 12288 number of nodes) for mesh interpolation. The results are shown in Figure 5(A). As the refinement level increases from 1 to 3, the MSE loss of the MIGN model exhibits a decline. For WDSP and MXSPD, the model achieves optimal performance at refinement level 4. In contrast, for the other four variables, the best performance is observed at refinement levels 3. From an empirical perspective, the optimal refinement level is typically chosen based on a mesh node count that is on the same order of magnitude or one order of magnitude lower than the number of station points.

**Mesh neighbors analysis** Figure 5(B) illustrates the MIGN perfomance across different mesh neighbor. We observe that using 10 neighbors yields the lowest loss for almost all variables. In contrast, performance significantly degrades when using only 2 neighbors due to limited information, and again when using 40 neighbors, likely due to the inclusion of distant or irrelevant nodes introducing noise.

## 4.6 Spherical Harmonics Degree Analysis

To evaluate the effectiveness of spherical harmonics, we conduct experiments with varying degrees of location embedding. The results are displayed in Table 5. We discover that, with the degree of spherical harmonics increasing from 0 to 2, MIGN achieves relatively better performance. For example, the MSE of the SLP and MXSPD decreases from 23.01/20.56 to 20.09/19.73. Because the rise of degree could make embedding approximate the higher-frequency harmonic, indicating a more precise representation of the location. When the degree increases from 2 to 3, the improvement in spherical harmonics embedding becomes marginal.

## 4.7 Empirical analysis

We visualize the global loss of MAX TEMP in Figure 6. The results reveal a significant regional variation in the difficulty of the prediction. For maximum temperature, inland areas of North America and northern Asia exhibit higher prediction errors compared to western Europe and Africa. STGCN and HD-TTS consistently show increased losses in both data-rich regions (e.g., North America) and data-scarce regions (e.g., northern Asia), indicating their limited ability to capture the underlying

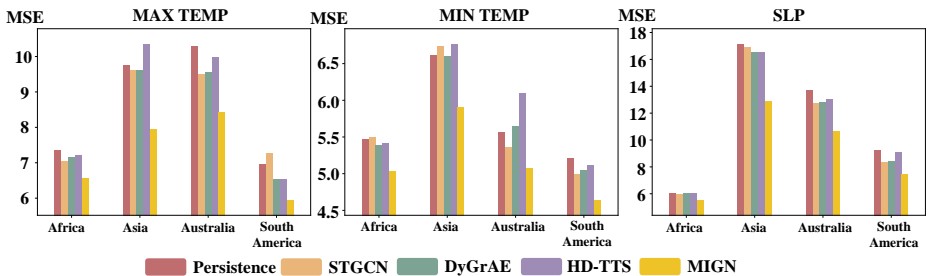

Figure 4: Comparison of different models in data-scarce regions.

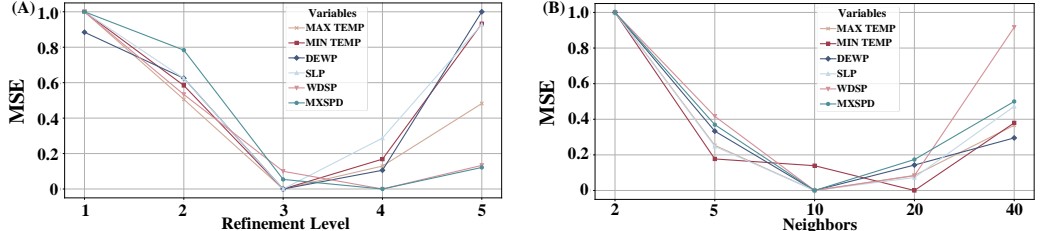

Figure 5: Comparison of model performance with different mesh hyperparameter settings

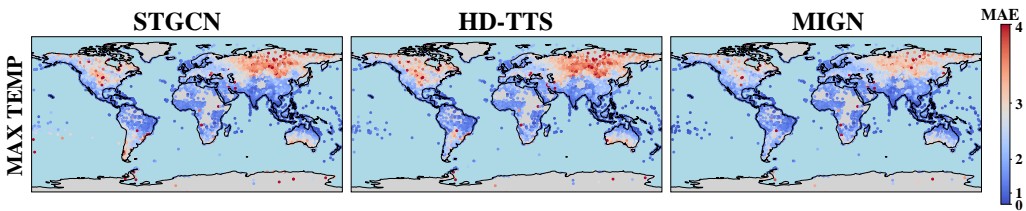

Figure 6: The global MAE distribution of MAX TEMP in testing set

patterns in these regions. In contrast, our model demonstrates superior performance, which indicates that the mesh design can capture the pattern of data-dense and data-sparse regions at the same time.

## 5 Conclusion and Future Works

In this work, we propose a MIGN framework for dynamic and spatially irregular global weather forecasting. It mitigates the spatially irregular problem by using mesh interpolation. We propose parametric spherical harmonics location embedding to learn the global weather information. Extensive experiments show that MIGN outperforms existing spatial-temporal models. Ablation studies demonstrate the effectiveness of the model designs and we further explored the hyperparameters in the mesh construction and the degree of spherical harmonics. Empirical analysis and generalization studies further illustrate the superior generalization ability. Due to the sparse distribution of weather stations over marine areas, our dataset primarily focuses on land-based observations. In future work, we plan to incorporate marine observation data to further enhance the robustness and generalization of our model in ocean-related scenarios.

**Limitations** Due to the sparse distribution of weather stations over marine areas, our dataset primarily focuses on land-based observations. However, incorporating additional data sources covering global oceans could further improve the performance of MIGN. Since the Earth operates as an interconnected system, integrating marine data would provide a more complete representation of global weather patterns.

## 6  Acknowledgement

This work is supported by the Guangzhou Industrial Information and Intelligent Key Laboratory Project (No. 2024A03J0628) and Guangdong S&T "1+1+1" Joint Funding Program C019.

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

# A Appendix

## A.1 Broader Impacts

Climate change has enhanced weather variability and extreme event frequency, such as heatwaves, droughts, and heavy rainfall, resulting in enormous socioeconomic loss. Accurate weather forecasting, especially in urban areas, is crucial for mitigating their impacts and benefiting various aspects of human life, including transportation management, agricultural planning, and resource allocation. Although multiple weather foundation models have been proposed, they focus on coarse-grained global forecasting of reanalysis data. Accurate predicting weather station observations, which are closer to urban areas and with fewer biases, are critical to weather forecasting applications.

## A.2 Limitations

Direct observation prediction models are inherently limited by the spatial distribution of stations. Unlike gridded models, which provide uniform coverage across the globe, station-based models can only make predictions at locations where observation data exists. Consequently, regions with sparse or no stations—such as remote oceans, deserts, or polar areas—remain unobserved and cannot be accurately modeled. This limitation restricts the ability of station-based approaches, especially when applied to areas far from the existing observational network.

## A.3 Additional Related Work

GINO [21] also leverages latent regular grid. However, GINO aims to simulate computational fluid dynamics. Such different application scopes result in distinctive model designs. GINO utilizes a regular 3D grid for variable input geometry. In contrast, MIGN employs a HEALPix mesh as the regular grid, which is aligned with the inherent spherical geometry of Earth. To model the spatially irregular and dynamic station distribution, we further incorporate a spherical harmonic embedding to enhance the spatial generalization ability of the model, which is not considered in GINO.

## A.4 Temporal Format of MIGN

MIGN can be naturally extended to a multi-step input–output setting. We define the input steps as a sequence of past observations from $t - n$ to $t$, and the output steps as the sequence of future predictions from $t + 1$ to $t + m$. Formally, the input consists of station nodes $\mathcal{V}_s^{t-n}, \ldots, \mathcal{V}_s^{t-1}, \mathcal{V}_s^t$ and their corresponding mesh nodes $\mathcal{V}_h^{t-n}, \ldots, \mathcal{V}_h^{t-1}, \mathcal{V}_h^t$.

For each input step, we independently apply the Mesh Interpolation Encoder and message passing as defined in Eq. (10)–(13), producing hidden states $\mathbf{h}_{v_h^{t-n}}^l, \ldots, \mathbf{h}_{v_h^t}^l$ for mesh nodes and aggregated processor states $\mathbf{H}_h^{t-n}, \ldots, \mathbf{H}_h^t$.

We then concatenate the temporal mesh representations $[\mathbf{H}_h^{t-n}; \ldots; \mathbf{H}_h^{t-1}; \mathbf{H}_h^t]$ and project them through a linear layer to obtain the output latent states $[\mathbf{H}_h^t; \ldots; \mathbf{H}_h^{t+m}]$ for the mesh. Finally, for each output step, the Station Interpolation Decoder (Eq. (14)) maps mesh states back to station predictions, yielding $\hat{Y}_s^{t+1}, \hat{Y}_s^{t+2}, \ldots, \hat{Y}_s^{t+m}$.

## A.5 Spherical Harmonics Location Embedding

The illustration of spherical harmonics location embedding is shown in Figure 7. We regard the weather information of the station nodes as a learnable spherical harmonics function. The spherical harmonics can be precomputed according to the coordinates and we learn the weight in the model directly.

## A.6 More Details on Datasets

**Data source** Global Surface Summary of the Day(https://www.ncei.noaa.gov/metadata/geoportal/rest/metadata/item/gov.noaa.ncdc:C00516/html) is derived from The Integrated Surface Hourly (ISH) dataset. The latest daily summary data are normally available 1-2 days after the date-time of the observations used in the daily summaries. The updated fre-

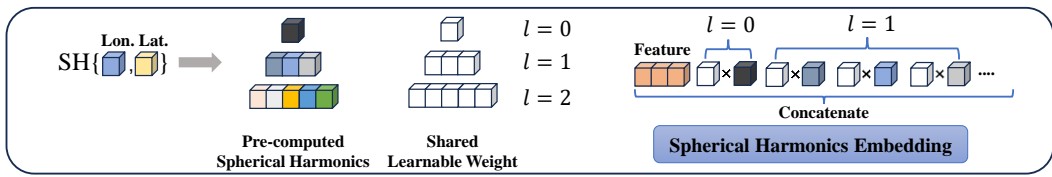

Figure 7: Spherical harmonics location embedding.

quency and reference time are daily and Greenwich Mean Time. The data can be download in the link `https://www.ncei.noaa.gov/data/global-summary-of-the-day/archive/`. We accessed the data on 2025/4/7. We computed both the total count of data points per variable over the period 2017–2024 in the Table 6.

**Dataset Selection Rationale**   We focus on daily maximum/minimum temperature and maximum sustained wind speed, which are inherently defined at daily timescales and not well captured by hourly aggregation. Daily forecasting also better aligns with medium-range horizons, whereas hourly data primarily serve short-term operations. Spatially, daily observations offer broader coverage, averaging over 10,000 stations per day compared to about 6,000 per hour (NOAA 2022), which enhances spatial pattern learning and supports global generalization. This expanded coverage provides a stronger foundation for capturing diverse geographical features and complex spatial dependencies essential to our modeling approach.

Table 6: Dataset variable statistic.

| VARIABLES | MAX TEMP | MIN TEMP | DEWP | SLP | WDSP | MXSPD |
|---|---|---|---|---|---|---|
| TOTAL NODE | 13260 | 13259 | 12709 | 9725 | 13014 | 12895 |

Table 7: The optimal training hyperparameter of baseline models for each variable.

| | | | | | | MAX TEMP | | | | | | |
|---|---|---|---|---|---|---|---|---|---|---|---|---|
| **Model** | STGCN | DyGrAE | STAR | GTN | MPNNLSTM | GPS | T&S-IMP | T&S-AMP | TTS-IMP | TTS-AMP | HD-TTS | ReDyNet | DualCast |
| **Learning rate** | 0.0086 | 0.0084 | 0.0033 | 0.0084 | 0.0040 | 0.0026 | 0.0059 | 0.0093 | 0.0044 | 0.0057 | 0.0007 | 0.0024 | 0.0035 |
| **Batch size** | 8 | 16 | 4 | 16 | 2 | 2 | 2 | 8 | 16 | 2 | 1 | 4 | 8 |
| **Hidden size** | 64 | 128 | 64 | 64 | 32 | 64 | 32 | 32 | 64 | 32 | 32 | 64 | 64 |
| | | | | | | **MIN TEMP** | | | | | | |
| **Model** | STGCN | DyGrAE | STAR | GTN | MPNNLSTM | GPS | T&S-IMP | T&S-AMP | TTS-IMP | TTS-AMP | HD-TTS | ReDyNet | DualCast |
| **Learning rate** | 0.0072 | 0.0053 | 0.0023 | 0.0034 | 0.0059 | 0.0076 | 0.0086 | 0.0067 | 0.0096 | 0.0087 | 0.0074 | 0.0063 | 0.0047 |
| **Batch size** | 16 | 16 | 16 | 4 | 4 | 4 | 2 | 8 | 8 | 1 | 2 | 8 | 8 |
| **Hidden size** | 32 | 128 | 64 | 128 | 32 | 32 | 64 | 64 | 32 | 64 | 64 | 32 | 64 |
| | | | | | | **DEWP** | | | | | | |
| **Model** | STGCN | DyGrAE | STAR | GTN | MPNNLSTM | GPS | T&S-IMP | T&S-AMP | TTS-IMP | TTS-AMP | HD-TTS | ReDyNet | DualCast |
| **Learning rate** | 0.0097 | 0.0032 | 0.0004 | 0.0065 | 0.0042 | 0.0005 | 0.0011 | 0.0071 | 0.0049 | 0.0004 | 0.0043 | 0.0017 | 0.0045 |
| **Batch size** | 16 | 8 | 4 | 8 | 16 | 16 | 2 | 8 | 16 | 2 | 1 | 4 | 4 |
| **Hidden size** | 32 | 128 | 64 | 64 | 64 | 32 | 128 | 128 | 128 | 32 | 128 | 64 | 128 |
| | | | | | | **SLP** | | | | | | |
| **Model** | STGCN | DyGrAE | STAR | GTN | MPNNLSTM | GPS | T&S-IMP | T&S-AMP | TTS-IMP | TTS-AMP | HD-TTS | ReDyNet | DualCast |
| **Learning rate** | 0.0059 | 0.0073 | 0.0065 | 0.0071 | 0.0038 | 0.0092 | 0.0063 | 0.0040 | 0.0059 | 0.0060 | 0.0044 | 0.0078 | 0.0024 |
| **Batch size** | 8 | 8 | 16 | 8 | 2 | 8 | 4 | 16 | 16 | 2 | 2 | 4 | 8 |
| **Hidden size** | 64 | 32 | 64 | 64 | 128 | 64 | 128 | 32 | 64 | 32 | 64 | 64 | 64 |
| | | | | | | **WDSP** | | | | | | |
| **Model** | STGCN | DyGrAE | STAR | GTN | MPNNLSTM | GPS | T&S-IMP | T&S-AMP | TTS-IMP | TTS-AMP | HD-TTS | ReDyNet | DualCast |
| **Learning rate** | 0.0004 | 0.0098 | 0.0082 | 0.0041 | 0.0027 | 0.0094 | 0.0061 | 0.0081 | 0.0045 | 0.0038 | 0.0012 | 0.0063 | 0.0034 |
| **Batch size** | 8 | 2 | 4 | 4 | 4 | 16 | 2 | 8 | 4 | 1 | 2 | 16 | 8 |
| **Hidden size** | 32 | 128 | 64 | 64 | 128 | 32 | 64 | 64 | 64 | 32 | 32 | 64 | 32 |
| | | | | | | **MXSPD** | | | | | | |
| **Model** | STGCN | DyGrAE | STAR | GTN | MPNNLSTM | GPS | T&S-IMP | T&S-AMP | TTS-IMP | TTS-AMP | HD-TTS | ReDyNet | DualCast |
| **Learning rate** | 0.0061 | 0.0045 | 0.0090 | 0.0023 | 0.0032 | 0.0018 | 0.0090 | 0.0042 | 0.0071 | 0.0023 | 0.0012 | 0.0035 | 0.0082 |
| **Batch size** | 4 | 8 | 4 | 2 | 2 | 2 | 1 | 2 | 16 | 1 | 2 | 4 | 8 |
| **Hidden size** | 32 | 64 | 64 | 64 | 32 | 32 | 64 | 64 | 32 | 64 | 32 | 64 | 64 |

## A.7 Baselines Implementation

- STGCN [41] is implemented base on Pytorch Geometric Temporal library `https://github.com/benedekrozemberczki/pytorch_geometric_temporal`. The graph convolution kernel size $K$ is set to 1.

- DyGrAE [9] is implemented base on Pytorch Geometric Temporal library `https://github.com/benedekrozemberczki/pytorch_geometric_temporal`. We use mean convolution aggregate method.

- STAR [42] is implemented base on Pytorch Geometric. The attention head is set to 4.

- GTN [33] is implemented base on Pytorch Geometric. The attention head is set to 4.

- GPS [31] is implemented base on Pytorch Geometric. The attention head is set to 1.

- MPNNLSTM [29] is implemented base on Pytorch Geometric Temporal library `https://github.com/benedekrozemberczki/pytorch_geometric_temporal`. The dropout rate is set to 0.5 and the Window is set to 1.

- T&S-IMP, T&S-AMP, TTS-IMP, TTS-AMP [5] are implemented base on official code `https://github.com/Graph-Machine-Learning-Group/taming-local-effects-stgnns`. We use 'elu' activation and mean normalization.

- HD-TTS [27] are implemented base on official code `https://github.com/marshka/hdtts`. We use anisoconv message passing method and kmis pooling method, with the dilation and kernel size are set to 1

- ReDyNet (2025) are implemented base on official code `https://github.com/wangzz-yyzz/ReDyNet`.

- DualCast (2025) are implemented base on official code `https://github.com/suzy0223/DualCast`.

We use Wandb Sweeps to automate hyperparameter search for each baseline and each varibales, utilizing the Bayesian sweep method. The hyperparameters are shown in Tabel7.

## A.8 Additional Results

### A.8.1 Time Horizon Analysis

**Input step analysis** To evaluate the models' ability to leverage varying lengths of historical input for accurate forecasting, we conduct an additional experiment using different input step settings. The results of MAX TEMP, MIN TEMP and DEWP, presented in Figure 8, demonstrate that MIGN consistently outperforms all other models across all three variables and input steps. While the baseline models show slight improvements as the number of input steps increases, MIGN achieves a more significant reduction in loss, particularly on DEWP, where the MSE drops from 8 to 7. Notably, increasing the input step from 3 to 4 yields only marginal gains for most models, indicating a diminishing return from longer input histories. The results of SLP, WDSP and MXSPD are presented in Figure 9. Across all variables, a general trend is observed where increasing the input step consistently leads to lower MSE loss, indicating that incorporating more historical information improves prediction accuracy. MIGN demonstrates the best overall performance. Its advantage becomes especially pronounced as the input step increases, achieving the lowest MSE in all three variables when the input step reaches 4. In contrast, Persistence, which serves as a naive baseline, maintains a high and constant error across all settings, emphasizing the benefits of using learning-based approaches.

**Autoregressive inference analysis** We evaluate the autoregressive forecasting performance of our model (MIGN) against a series of competitive baselines. Results are reported in Tables 8 and 9. Across all variables, MIGN consistently achieves the best performance in terms of both MSE and MAE. In particular, for MAX TEMP, MIGN reduces the total MAE to 2.72, outperforming the strongest baseline (STGCN, 2.96). For MIN TEMP, MIGN achieves a total MAE of 2.63, significantly lower than previous methods. Similar improvements are observed for DEWP, where MIGN yields a total MAE of 2.76, and for SLP, where MIGN achieves a total MAE of 4.36.

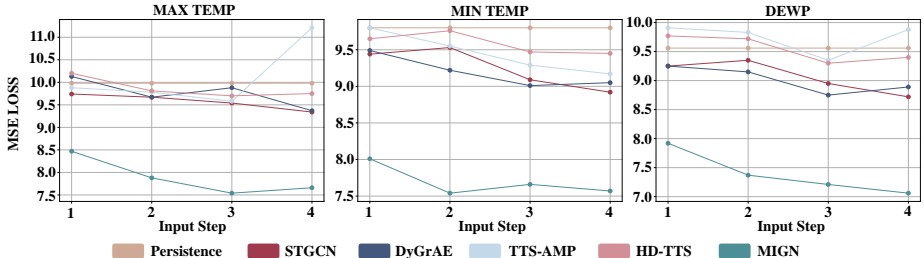

Figure 8: Comparison of different input steps on three key variables: MAX TEMP, MIN TEMP, and DEWP. MIGN achieves the best performance.

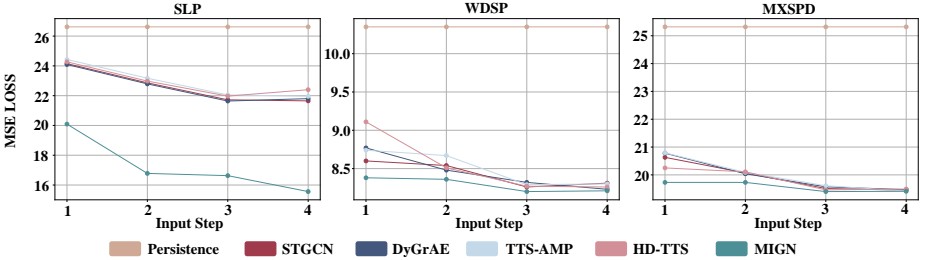

Figure 9: Comparison of different input steps on three key variables: SLP, DEWP, and MXSPD. MIGN achieves the best performance.

Notably, the performance gain becomes more pronounced at longer forecasting horizons (Step 2 and Step 3), indicating that MIGN is particularly effective at capturing temporal dependencies in extended autoregressive prediction. These results demonstrate that integrating mesh–station interactions enables our model to generalize better across different meteorological variables and forecasting horizons, thereby enhancing both short-term and long-term prediction accuracy.

Table 8: Bold font indicates the best result and Underline is the strongest baseline.

| | MAX TEMP($K$) | | | | | | | | MIN TEMP ($K$) | | | | | | | |
|---|---|---|---|---|---|---|---|---|---|---|---|---|---|---|---|---|
| Model | Step1 | | Step2 | | Step3 | | Total | | Step1 | | Step2 | | Step3 | | Total | |
| | MSE | MAE | MSE | MAE | MSE | MAE | MSE | MAE | MSE | MAE | MSE | MAE | MSE | MAE | MSE | MAE |
| STGCN (2017) | 9.74 | 2.22 | 17.75 | 3.11 | 22.55 | 3.54 | 16.68 | 2.96 | 9.44 | 2.11 | 17.21 | 2.98 | 21.32 | 3.36 | 16.04 | 2.83 |
| DyGrAE (2019) | 10.31 | 2.24 | 18.68 | 3.16 | 24.27 | 3.63 | 17.56 | 3.11 | 9.49 | 2.11 | 17.81 | 3.01 | 22.56 | 3.41 | 16.75 | 2.86 |
| GTN (2020) | 9.88 | 2.22 | 18.40 | 3.14 | 23.72 | 3.61 | 17.34 | 2.99 | 9.49 | 2.14 | 17.43 | 3.03 | 21.95 | 3.45 | 16.37 | 2.88 |
| T&S-IMP (2023) | 12.12 | 2.51 | 20.84 | 3.42 | 27.47 | 3.95 | 20.25 | 3.31 | 10.92 | 2.33 | 18.57 | 3.15 | 23.86 | 3.61 | 17.91 | 3.05 |
| T&S-AMP (2023) | 10.16 | 2.28 | 18.74 | 3.18 | 24.48 | 3.69 | 17.76 | 3.04 | 12.90 | 2.59 | 31.31 | 4.15 | 44.89 | 4.96 | 30.57 | 3.99 |
| TTS-IMP (2023) | 10.40 | 2.32 | 19.34 | 3.27 | 25.22 | 3.79 | 18.31 | 3.12 | 11.58 | 2.44 | 26.69 | 3.78 | 38.75 | 4.61 | 25.91 | 3.65 |
| TTS-AMP (2023) | 9.88 | 2.25 | 19.06 | 3.23 | 24.96 | 3.73 | 18.05 | 3.07 | 9.80 | 2.18 | 18.02 | 3.10 | 23.12 | 3.58 | 17.03 | 2.96 |
| HD-TTS (2024) | 10.20 | 2.33 | 18.64 | 3.27 | 24.18 | 3.78 | 17.67 | 3.12 | 9.65 | 2.17 | 17.13 | 3.02 | 21.35 | 3.43 | 16.07 | 2.88 |
| MIGN | **8.47** | **2.09** | **15.08** | **2.84** | **19.28** | **3.24** | **14.33** | **2.72** | **8.01** | **1.99** | **14.63** | **2.75** | **18.18** | **3.09** | **13.83** | **2.63** |

### A.8.2 Further Ablation study of location encoding

To investigate the impact of different location embedding strategies, we compare our proposed Spherical Harmonics (SH) embedding method with three commonly used coordinate-based approaches: **Direct Coordinate**, **WRAP**, and **Cartesian 3D**. The formulations of these methods are as follows. Let longitude $\lambda \in [-\pi, \pi]$ and latitude $\theta \in [-\pi/2, \pi/2]$.

- **Direct Coordinate:**

$$PE(\lambda, \theta) = (\lambda, \theta) \tag{15}$$

- **WRAP:**

$$PE(\lambda, \theta) = [\cos \lambda, \sin \lambda, \cos \theta, \sin \theta] \tag{16}$$

Table 9: Bold font indicates the best result and Underline is the strongest baseline.

| Model | DEWP($K$) Step1 | | Step2 | | Step3 | | Total | | SLP ($mb$) Step1 | | Step2 | | Step3 | | Total | |
|---|---|---|---|---|---|---|---|---|---|---|---|---|---|---|---|---|
| | MSE | MAE | MSE | MAE | MSE | MAE | MSE | MAE | MSE | MAE | MSE | MAE | MSE | MAE | MSE | MAE |
| STGCN (2017) | 9.25 | 2.11 | 18.59 | 3.07 | 23.13 | 3.49 | 17.08 | 2.90 | 24.15 | 3.42 | 46.24 | 4.85 | 55.79 | 5.38 | 42.03 | 4.55 |
| DyGrAE (2019) | 9.25 | 2.10 | 18.68 | 3.05 | 23.18 | 3.45 | 17.12 | 2.88 | 24.09 | 3.40 | 45.99 | 4.81 | 55.11 | 5.30 | 41.71 | 4.51 |
| GTN (2020) | 9.51 | 2.16 | 19.66 | 3.19 | 25.25 | 3.68 | 18.22 | 3.02 | 24.49 | 3.44 | 47.57 | 4.93 | 57.31 | 5.47 | 43.11 | 4.61 |
| T&S-IMP (2023) | 10.80 | 2.31 | 19.87 | 3.22 | 25.26 | 3.69 | 18.67 | 3.08 | 24.70 | 3.46 | 46.39 | 4.88 | 55.75 | 5.40 | 42.31 | 4.59 |
| T&S-AMP (2023) | 9.43 | 2.16 | 19.25 | 3.18 | 24.35 | 3.65 | 17.80 | 3.02 | 24.38 | 3.44 | 45.59 | 4.82 | 54.66 | 5.31 | 41.47 | 4.52 |
| TTS-IMP (2023) | 13.69 | 2.64 | 26.89 | 3.79 | 46.86 | 4.62 | 28.57 | 3.64 | 24.76 | 3.47 | 46.97 | 4.91 | 57.41 | 5.48 | 43.04 | 4.62 |
| TTS-AMP (2023) | 9.91 | 2.19 | 20.28 | 3.21 | 26.09 | 3.70 | 18.83 | 3.04 | 24.43 | 3.45 | 46.43 | 4.89 | 55.84 | 5.40 | 42.22 | 4.58 |
| HD-TTS (2024) | 9.77 | 2.21 | 19.35 | 3.20 | 24.54 | 3.69 | 17.93 | 3.04 | 24.27 | 3.44 | 46.68 | 4.86 | 56.12 | 5.38 | 42.35 | 4.56 |
| MIGN | **7.92** | **1.97** | **16.35** | **2.90** | **20.45** | **3.31** | **15.21** | **2.76** | **20.09** | **3.14** | **43.29** | **4.64** | **55.29** | **5.26** | **39.93** | **4.36** |

- **Cartesian 3D:**

$$PE(\lambda, \theta) = [\cos\theta\cos\lambda, \cos\theta\sin\lambda, \sin\theta] \tag{17}$$

We apply the above three location embeddings as well as our SH embedding method to both the encoder and decoder of our model. Table 10 reports the mean squared error (MSE) on six meteorological variables. Our proposed SH embedding consistently outperforms the baseline methods, highlighting the effectiveness of modeling spherical positional information using harmonics.

Table 10: Comparison of different location embeddings in terms of mean squared error (MSE). Lower values indicate better performance.

| Model Variant | MAX TEMP | MIN TEMP | DEWP | SLP | WDSP | MXSPD |
|---|---|---|---|---|---|---|
| W/O | 9.04±0.08 | 8.71±0.05 | 8.71±0.04 | 23.01±0.07 | 8.76±0.02 | 20.63±0.04 |
| Direct | 8.88±0.06 | 8.57±0.09 | 8.35±0.06 | 21.89±0.06 | 8.64±0.01 | 19.85±0.04 |
| WRAP | 8.70±0.05 | 8.32±0.08 | 8.23±0.03 | 21.14±0.05 | 8.52±0.03 | 19.83±0.08 |
| Cartesian 3D | 8.67±0.09 | 8.34±0.05 | 8.19±0.03 | 21.21±0.05 | 8.48±0.03 | 19.86±0.06 |
| **SH Embedding** | **8.47±0.05** | **8.01±0.04** | **7.92±0.05** | **20.09±0.07** | **8.38±0.01** | **19.73±0.05** |

### A.8.3 Sparse region analysis

To assess the generalization ability of the models in data-scarce regions, we conduct experiments across Africa, Asia, Australia, and South America, focusing on three key meteorological variables: DEWP, WDSP, and MXSPD, as shown in Figure 11. Across all variables and regions, MIGN consistently achieves the lowest MSE, demonstrating robust performance in regions with limited observational data. Particularly in South America, MIGN significantly outperforms other baselines for DEWP, achieving an MSE below 3.5, while other models yield considerably higher errors. Similarly, for WDSP and MXSPD, MIGN maintains stable and low error rates across all continents, showcasing its ability to generalize well across diverse climatic conditions. These results further confirm MIGN's effectiveness in learning reliable patterns even when data availability is limited.

### A.8.4 Mesh analysis

The MAE result of refinement level analysis and mesh neighbors analysis is shown in Figure 12. In Figure 12(A), performance improves as the refinement level increases from 1 to 3, reaching the lowest MAE at level 3. Beyond this point, error increases again, suggesting that too fine a mesh may introduce noise. In Figure 12(B), the optimal number of neighbors is around 5–10, where MAE is minimized. Too few neighbors (e.g., 2) lack spatial context, while too many (e.g., 20 or 40) may introduce irrelevant information, hurting performance.

### A.8.5 Visualization of Spherical Harmonics embedding

Since our spherical harmonics are designed to learn the coefficients $w_n^m$, we compute the spherical function $f(\lambda, \phi) = \sum_{n=0}^{3}\sum_{m=-n}^{n} w_n^m Y_n^m(\lambda, \phi)$, where $w_n^m$ are the learned coefficients in the MAX task with degree 3. As illustrated in the Figure 10, different regions on the globe exhibit distinct colors: North America appears purple, South Africa black, Europe yellow, and Asia red. This indicates that the spherical harmonics embedding can capture location-specific information.

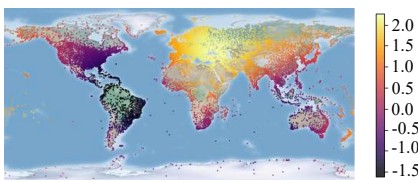

Figure 10: The visualization of learned spherical harmonics embedding.

### A.8.6 Comparison with gridded model

**Grid-to-Station Evaluation** To further examine the necessity of station-based approaches, we conducted a direct comparison against gridded reanalysis models. Specifically, we evaluated Pangu's 2022 gridded forecasts from WeatherBench2 by bilinearly interpolating them to station locations, ensuring a fair comparison with our station-based model (MIGN). MIGN was trained on station observations from 2017–2020, validated on 2021, and tested on 2022. Results are summarized in Table 11:

Table 11: Comparison with gridded model

| VARIABLES | MAX | MIN | WDSP |
|---|---|---|---|
| PANGU(0.25° RESOLUTION) | 10.84 | 9.95 | 9.76 |
| MIGN | 8.71 | 9.02 | 8.60 |

Despite reanalysis data offering broader spatial coverage and higher temporal frequency, MIGN consistently outperforms Pangu across all three variables when evaluated at station locations. The advantages of station-based learning are threefold: (i) it directly leverages ground-truth observations without inheriting potential biases or smoothing artifacts introduced during reanalysis assimilation, (ii) it preserves fine-scale spatial variability essential for capturing extremes and urban microclimates, and (iii) it avoids the computational burden of handling full 4D gridded datasets, enabling efficient training and deployment even under limited hardware resources.

These strengths make station-based approaches particularly valuable in scenarios where high-quality observations are available, offering sharper local accuracy and practical efficiency that complement the broad coverage of reanalysis methods.

**Unmonitored-Location Generalization** To further investigate the spatial generalization of station-based methods, we conducted an experiment where 10% of observation sites were withheld as unmonitored ground truth points. The remaining 90% of stations were used for training our MIGN model. For a fair comparison, both Pangu and MIGN predictions were bilinearly interpolated to these unmonitored sites. Results are summarized in Table 12:

Table 12: Comparison with gridded model

| VARIABLES | MAX | MIN | WDSP |
|---|---|---|---|
| PANGU(0.25° RESOLUTION) | 11.45 | 10.45 | 9.98 |
| PANGU(1° RESOLUTION) | 14.12 | 13.25 | 12.87 |
| MIGN | 13.84 | 12.87 | 13.14 |

MIGN outperforms Pangu at 1.00° resolution on MAX TEMP and MIN TEMP, and achieves competitive WDSP accuracy. However, Pangu at 0.25° grid spacing remains superior, which can be attributed to its massive data and computational advantages. Pangu is trained on over one million global reanalysis points per snapshot, leveraging rich multi-variable inputs (geopotential height, temperature, humidity, wind, etc.) across multiple vertical levels, requiring more than 200 TB of training data. By contrast, MIGN operates with only 10,000 data points per day and a total

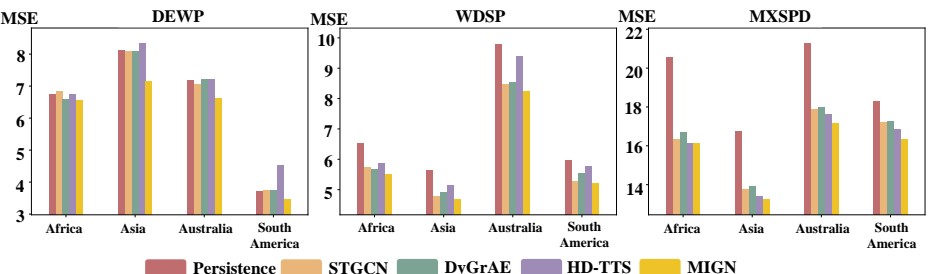

Figure 11: Comparison of different models in data-scarce regions (Africa, Asia, Australia, South America) on three key variables: DEWP, WDSP, and MXSPD.

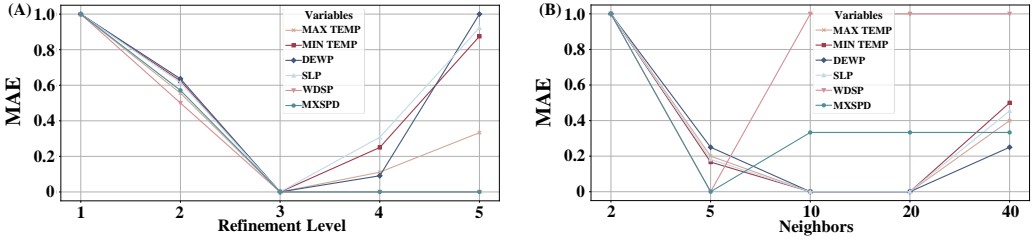

(a) Comparison of model performance with different refinement level

(b) Comparison of model performance with different neighbors

Figure 12: Comparison of model performance with different mesh hyperparameter settings

data volume of roughly 10 GB, highlighting its efficiency and accessibility under data-scarce or resource-constrained conditions.

## A.9 Computational Cost Analysis

**Analysis of mesh interpolation** Mesh interpolation is achieved by constructing a nearest neighbors graph between feature nodes and Healpix nodes. Suppose there are $M$ feature nodes and $N$ Heslpix nodes. Using a brute-force approach, the computational complexity is $O(MN)$. For baselines, the computational complexity of nearest neighbor connections among the feature nodes is $O(M^2)$. Mesh analysis experiments Figure 12a indicate that the optimal number of Healpix nodes $N$ is smaller than the number of feature nodes $M$, meaning that $N < M$, thus $O(MN) < O(M^2)$. Furthermore, by utilizing a KD-Tree algorithm, we can further reduce the complexity from $O(MN)$ to $O(NlogN + MlogN)$.

**Training and inference efficiency** To further demonstrate the training and inference efficiency of our model, we compare the training and inference times per step of several baselines with MIGN on an NVIDIA RTX 3090 GPU, as shown in the following table. We observe that the training and inference time of MIGN is comparable to that of STGCN and MPNNLSTM, demonstrating its efficiency and practical effectiveness.

**Analysis of spherical harmonics** Spherical harmonic basis functions are precomputed and stored for efficiency. The Spherical Harmonics (SH) Degree Analysis Experiment 5 demonstrates that a degree of 2 is sufficient for location embedding. Thus, its computational cost is linear to the node number, which is not the primary time-consuming component. We measure the processing speed for MAX TEMP data with a degree-3 SH embedding (13260 nodes) on an AMD EPYC 75F3 32-Core Processor. The computation completes in just 2s.

Table 13: Training and inference time per step for different models.

| Model | STGCN | TGCN | DyGrAE | MPNNLSTM | GPS | HD-TTS | MIGN |
|---|---|---|---|---|---|---|---|
| Training time per step (s) | 0.013 | 0.014 | 0.016 | 0.012 | 0.048 | 3.25 | 0.013 |
| Inference time per step (s) | 0.004 | 0.010 | 0.012 | 0.011 | 0.019 | 3.03 | 0.006 |

## A.10 Empirical analysis

To further illustrate our motivation, we visualize the global loss of MAX TEMP, MIN TEMP, DEWP, SLP in Figure 13, 14, 15, 16. The results reveal that prediction difficulty varies significantly across regions. For MAX TEMP, MIN TEMP and DEWP, inland areas of North America and northern Asia exhibit higher prediction errors compared to Western Europe and Africa, highlighting distinct regional characteristics and suggesting that different regions follow different weather patterns. Baseline models consistently show higher losses in North America and northern Asia, indicating their limited ability to capture the underlying patterns in these regions. In contrast, our model demonstrates superior performance across both the United States and the Asian continent. This suggests that mesh interpolation and spherical harmonics facilitate the learning of global patterns and effectively capture regional features. For SLP, model performance tends to degrade in high-latitude regions, including Western Europe and North America. This pattern suggests increased difficulty in capturing surface-level pressure dynamics in these areas, possibly due to more complex atmospheric interactions and variability at higher latitudes. Baseline models exhibit particularly high prediction errors in these regions, reinforcing their limitations in modeling such complexity. In comparison, our model maintains relatively stable performance, indicating its enhanced capacity to learn intricate spatial patterns through mesh interpolation and spherical harmonics.

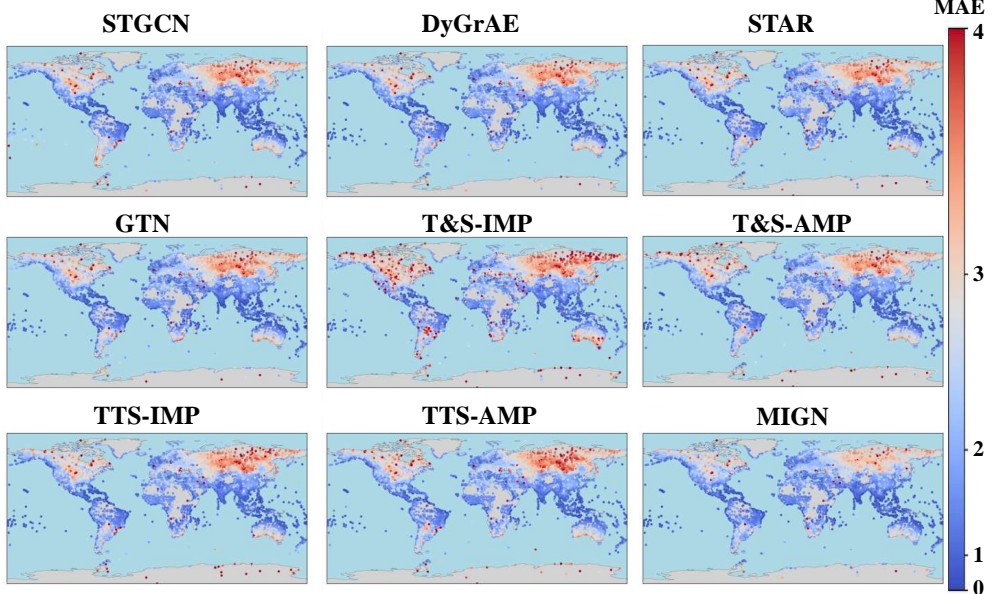

Figure 13: The global MAE distribution of MAX TEMP in testing set

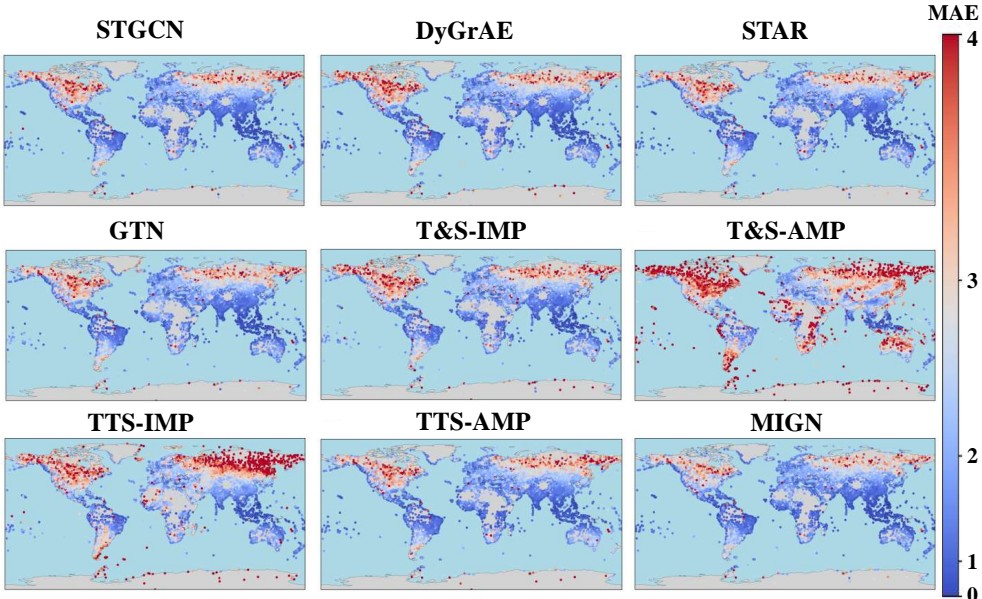

Figure 14: The global MAE distribution of MIN TEMP in testing set

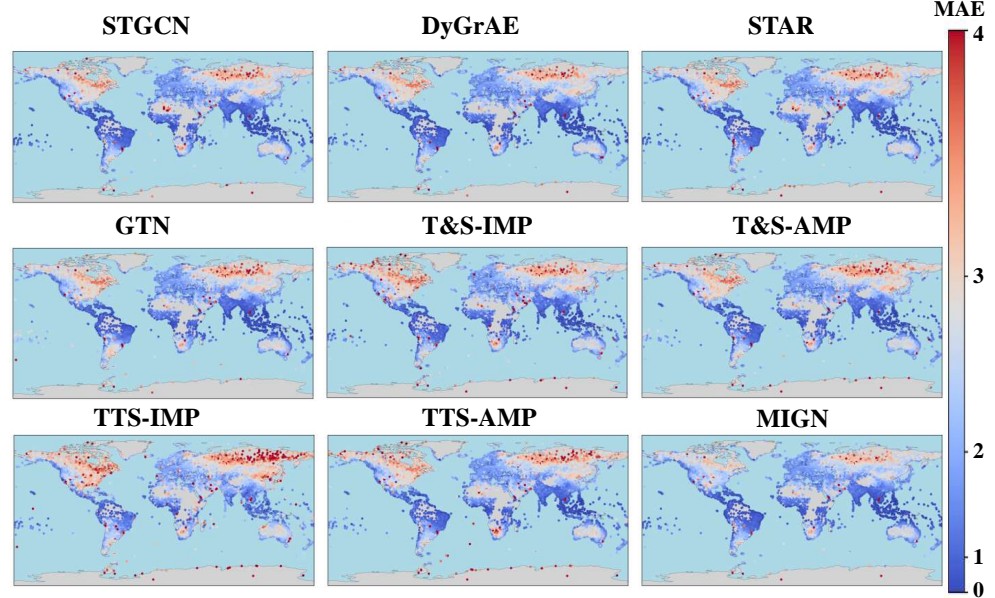

Figure 15: The global MAE distribution of DEWP in testing set

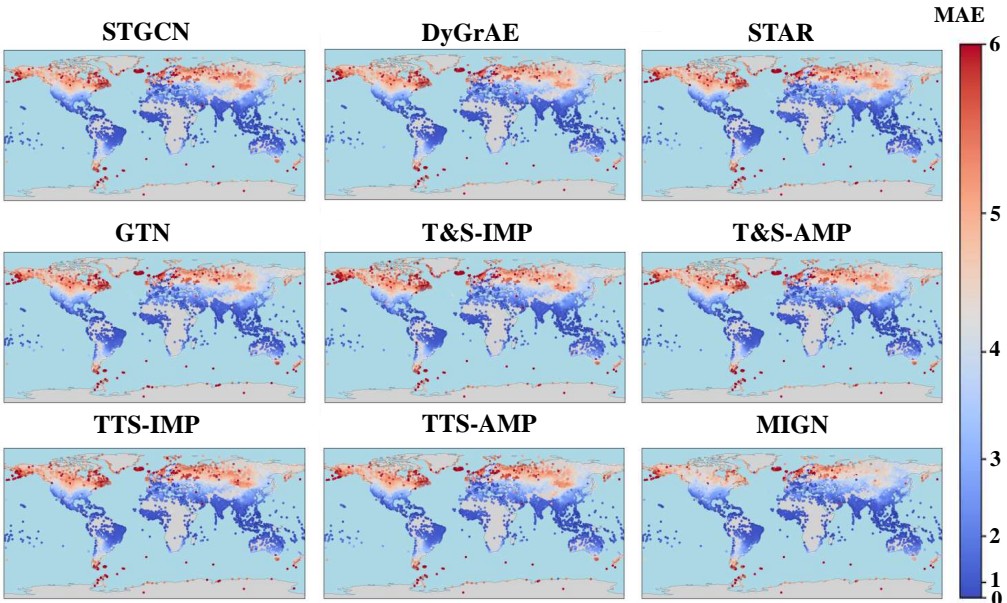

Figure 16: The global MAE distribution of SLP in testing set

