# OpenReview forum: "Mesh Interpolation Graph Network for Dynamic and Spatially Irregular Global Weather Forecasting"
_NeurIPS.cc/2025/Conference — NeurIPS 2025 poster_

### Official Review · Reviewer_yq1V · 2025-07-01

**Clarity:** 3
**Significance:** 2
**Originality:** 2
**Rating:** 4
**Confidence:** 4

**Summary:**

The paper proposes MIGN, a deep learning model for weather forecasting that handles irregularly distributed sensor data, different from previous work, which typically assumes input data is arranged on a regular grid. Sensors are also allowed to be missing for some time steps. The model encodes sensor observations onto a global mesh, performs message passing across the mesh, and decodes the results back to the sensor locations. A key design element is the use of spherical harmonics as positional encodings to improve spatial generalization. Experimental results show that MIGN outperforms prior graph-based approaches on the considered task.

**Questions:**

1. Why did the authors choose to evaluate only on a dataset with daily resolution, and why is the forecasting task limited to one-day-ahead predictions? Given the smoothness of daily-resolution data, this setup seems to trivialize the forecasting task. Could the authors clarify whether MIGN could be tested on higher-frequency data (e.g., hourly), where forecasting would be more challenging and meaningful?
2. Why is the input limited to a single time step? Results in the appendix suggest that increasing the look-back window improves performance across all models, including MIGN. Could the authors explain the rationale for restricting the input window to one step in the main experiments?
3. How are node-specific parameters handled in global-local architectures under a dynamic sensor network? In scenarios where sensors appear only at test time, node-specific parameters (which are trained end-to-end) remain randomly initialized. How is this issue addressed, particularly in the Global Generalization Analysis, where many sensors are unseen during training?
4. How does MIGN handle longer input sequences, and how do the baselines handle missing stations in those cases? For example, in a longer input window, how is MIGN modified to integrate multiple time steps? For the baselines, how is prediction handled when a sensor is only observed at time t+1, but not at time t? Which graph is used for the baselines?
5. How were the baselines evaluated on this dataset, and do they share a unified codebase for training and evaluation? The code repository only includes MIGN. To ensure a fair comparison, it's important to understand whether all baselines were implemented and evaluated under consistent experimental conditions. Could the authors provide more details on the training and evaluation protocols used for the baselines?

**Ethical Concerns:**

["NO or VERY MINOR ethics concerns only"]

**Final Justification:**

Following the rebuttal and further discussion, I have revised my recommendation to borderline accept. The authors have provided thoughtful clarifications and additional results that address several key concerns, particularly regarding the task formulation and experimental design. My final evaluation is based on the following considerations:

* **Resolved issues**

  * The authors clarified the rationale behind using daily-resolution data and justified the forecasting setup. The additional multi-step forecasting results are more aligned with common practices and significantly strengthen the paper's relevance.
  * The authors addressed concerns regarding implementation details (e.g., temporal extension, batch sizes, and handling of spatial embeddings in baselines). These clarifications improve understanding of the method and confidence in the fairness of the comparisons.

* **Remaining concerns**

  * The evaluation remains limited to a dataset with relatively coarse temporal resolution, which may reduce the generalization and impact of results. However, this is mitigated by the challenging nature of the dynamic sensor setting considered.
  * While improved in the rebuttal, the manuscript would benefit from a thorough revision to improve clarity, organization, and notation in the final version.

Overall, the paper addresses a relevant and timely problem, introduces a novel and interesting approach, and demonstrates promising performance. While certain limitations remain, the authors' responses and proposed revisions sufficiently alleviate my initial concerns, warranting a borderline accept recommendation.

**Limitations:**

While the authors do mention some limitations, two important points deserve more attention:

1. The forecasting task focuses on daily summary weather prediction, which, due to its low temporal resolution, may not fully capture the complexity or practical demands of real-world weather forecasting applications.
2. The setup of using only the previous day's data to predict the next day imposes a restrictive constraint on forecasting, and is especially limiting for sequence models.

These limitations should be more explicitly acknowledged and discussed in the manuscript.

**Paper Formatting Concerns:**

Some formatting issues are present. Spacing around section headings is inconsistent, especially noticeable in Section 5. Equations, such as Eq. 9, appear to have been manually modified in terms of spacing or layout. Section titles alternate between title case and sentence case.

**Quality:**

2

**Strengths And Weaknesses:**

## Strengths

1. The paper tackles a highly relevant and timely task (weather forecasting) and contributes to the broader effort to reduce the computational cost associated with traditional numerical weather prediction (NWP) models.
2. The setting adopted in this work falls within the Direct Observation Prediction (DOP) paradigm, which focuses on forecasting weather directly from meteorological observations. This is a particularly challenging setup due to the spatial sparsity of the data and the dynamic nature of the sensor network (i.e., sensors may appear or disappear). This makes the problem formulation both realistic and impactful.
3. While there is room for improvement in the clarity of the text and notation, the core ideas of the proposed approach are communicated effectively. In particular, the figures are well-designed and convey the intended messages clearly and concisely.
4. The paper includes a wide experimental analysis, comparing MIGN against both standard baselines and state-of-the-art spatiotemporal graph neural networks. Ablation studies are thoughtfully conducted and demonstrate the contribution of each architectural component. Additionally, confidence intervals are reported, which strengthens the reliability of the results.

## Weaknesses

1. The proposed approach is evaluated on a single dataset, derived from the NOAA Global Surface Summary of the Day service. As the name suggests, this dataset contains daily-resolution data, resulting in very smooth time series. Furthermore, the chosen task — predicting one day ahead — is of limited interest in this context (compared to, e.g., hourly data). Given the coarse temporal resolution, such forecasting is often trivial, as evidenced by the strong performance of the persistence model (which simply copies the previous day's value), which frequently ranks among the best-performing baselines.
2. The problem is set as next-day prediction based solely on data from the previous day. This setup imposes an overly restrictive condition on sequence-processing models, such as those used in several baselines. Moreover, figures 8 and 9 in the appendix show that increasing the look-back window improves performance across all models, including MIGN. Therefore, it is unclear why the main experiments in the paper are limited to a one-step history.
3. Most of the baselines rely on recurrent neural networks (RNNs) to model temporal dynamics. These models are inherently designed to process sequences, and are thus disadvantaged when constrained to a single static input matrix, as is the case here. In contrast, MIGN operates on static graph structures (see Equation 9), which gives it a natural advantage under this setting.
4. Many of the baselines, particularly those labeled “global-local” and the HD-TTS model, are equipped with node-specific parameters. This design choice introduces new challenges in scenarios with dynamic sensor networks. Since these parameters are learned during training, sensors not seen in the training set retain randomly initialized weights. As a result, performance may degrade significantly when many sensors are encountered for the first time during testing or appear sporadically in the training set. This issue is particularly relevant for the Global Generalization Analysis, in which half the sensors are held out during training. The paper should explicitly address how this challenge is handled.
5. The code repository provided with the paper includes only the implementation of the proposed architecture. Precise experimental details for the baselines are not specified, either in the repository or in the paper. This omission prevents a full understanding of how experiments were conducted and undermines reproducibility. Additionally, several baselines were trained using very small batch sizes (1–2 samples), which can significantly hinder their performance.
6. While the core ideas are conveyed adequately, the paper suffers from issues in organization, notation, and language. Some symbols and operations are not properly introduced (e.g., “[ ; ; ]”), and Section 3.3 references results and acronyms from the following section without first describing the experimental setup. Encoding and decoding procedures are initially described without spherical harmonics in Section 3.1, only to be redefined with harmonics in Section 3.2, creating confusion. The use of "would" in lines 158–162 is also grammatically incorrect. These are just a few examples, but overall, the manuscript would greatly benefit from a thorough revision.

---

> ### Author Rebuttal · Authors · 2025-07-31
>
> We sincerely thank the reviewer for listing the strengths. After carefully reading the review, we are afraid that the reviewer significantly **Underestimates** the problem challenges. We have tried our best to eliminate the misunderstanding as follows.  If you are satisfied with the response, we kindly request your approval by considering an improved score.
>
> > Weakness 1 & Questions 1 & Limitation 1: Daily-resolution data is very smooth.
>
> Thanks for your suggestion. We would like to clarify that the daily data is **NOT** smoother than the hourly data in our one-day head prediction setting, especially for the extreme variables (i.e., daily maximum, minimum temperature, and maximum sustained wind speed). To verify it, we compared the Mean Absolute First Difference (MAFD) of the hourly records in NOAA with their corresponding NOAA Global Daily station data. Due to time constraints, we were only able to download a randomly selected subset of 1,000 stations from the NOAA Global Hourly dataset. While this is a limited sample, we are in the process of downloading more hourly data to support a broader comparison. The current results are as follows:
>
> Mean Hourly Fluctuations
> |Dataset|TEMP|DEWP|WDSP|
> |-------------|--------|--------|--------|
> |Hourly|1.44 °C|0.936 °C |0.96m/s |
>
> Mean Daily Fluctuations
> |Dataset|MEAN TEMP|MAX TEMP|MIN TEMP|DEWP|WDSP|MXSPD|
> |-------------|--------|--------|--------|--------|--------|--------|
> |Daily|1.96°C|2.40 °C|2.35°C| 2.22 °C|1.14m/s | 1.76m/s|
>
> Our analysis reveals that the daily time series exhibits significant non-smooth characteristics, which present substantial challenges for one-day-ahead prediction tasks. The relatively strong performance of the Persistence baseline in this context should not be interpreted as indicating task triviality. Rather, it demonstrates the inherent difficulty in capturing meaningful temporal patterns from daily-scale weather data. It highlights the need for specialized architectural designs (i.e., MIGN) tailored to daily prediction scenarios.
>
> We appreciate the reviewer’s insightful suggestion regarding higher-frequency data. While MIGN could theoretically be extended to hourly forecasting, our current work focuses on daily resolution due to its greater relevance to our target application. Specifically, station configurations in hourly settings often exhibit less dynamic variability, whereas the daily scale better captures the spatial-temporal patterns critical to our study. Future work could explore adapting MIGN to finer temporal resolutions, but we argue that daily prediction remains a practically significant and scientifically valuable benchmark for our proposed framework.
>
> > Weaknesses 2 & Questions 2: Why are the main experiments in the paper limited to a one-step history?
>
> Thanks for the comment. While our primary focus in this work is on demonstrating MIGN's spatial generalization capabilities, we acknowledge the importance of investigating multi-step historical inputs. Due to space constraints in the original submission, we initially presented only single-step history results. As we have shown in Figures 8 and 9 in the Appendix, MIGN maintains superior performance even with extended input histories. The illustration would not affect our contribution.
>
> We acknowledge the importance of long input history analysis and have included these results in the revised manuscript's main text.
>
> > Weaknesses 3 & Questions 3 & Limitations 2: MIGN operates on static graph structures, which gives it a natural advantage under this setting.
>
> Thanks for the question. As we have shown in Fig.8 and Fig.9, MIGN outperforms baselines even if increasing the length of history.
>
> > Weaknesses 4 & Questions 3: How are node-specific parameters handled in global-local architectures under a dynamic sensor network?
>
> Thank you for your question. Using node-specific parameters deteriorates performance in our dynamic setting, as the test set contains nodes that are not seen during training. To ensure fair comparison and enable generalization to unseen nodes, we replace node-specific parameters with direct position embeddings, defined as $PE(\lambda, \theta) = (\lambda, \theta)$。 This allows the models to retain spatial awareness without overfitting to specific nodes.
>
> > Weaknesses 5 & Questions 5: Experimental details.
>
>
> We appreciate the reviewer's inquiry regarding implementation details. For comprehensive information about baseline implementations, please refer to Appendix A.5. Regarding hyperparameter optimization, we employed a rigorous Bayesian search approach using Weights & Biases (WandB) rather than manual tuning, as documented in Table 6 (Appendix A.5). This systematic process ensured optimal configurations for all models.
>
> Our hyperparameter search space included:
> * Hidden sizes: [8,16,32,64,128]
> * Batch sizes: [1,2,4,8,16,32] (limited to [1,2] for memory-intensive models like HDTTS)
> * Learning rates: $1\times 10^{-4}$ to $1\times 10^{-2}$
>
> The final reported results reflect the best-performing configurations identified through this exhaustive search, with some models achieving optimal performance at smaller batch sizes (1-2). This methodology guarantees fair and reproducible comparisons across all evaluated approaches.
>
> > Weaknesses 6: Presentation.
>
> We sincerely appreciate the reviewer's careful reading and apologize for the typographical errors in our current draft. We will conduct thorough proofreading to ensure all notations are consistent and the manuscript's readability is significantly improved in the revised version.
>
> > Questions 4: How does MIGN handle longer input sequences, and how do the baselines handle missing stations in those cases?
>
> Thanks for the question. For different input time steps, station features are passed to the corresponding Healpix nodes through message passing. An MLP is then used to map the features from the encoder's Healpix nodes to those in the decoder. For the baselines, we follow the approach used in HD-TTS: if a sensor (node) is not observed at time t, but appears at t+1, it is treated as missing data at t and initialized with the mean value. This allows the construction of a complete graph at each timestep.
>
> > Questions 5: How were the baselines evaluated on this dataset, and do they share a unified codebase for training and evaluation?
>
> All baseline models were implemented, trained, and evaluated using a unified PyTorch Lightning framework to ensure consistency with MIGN's experimental setup. We use the Adam optimizer with an early stopping strategy (patience = 3). For graph construction in the baselines, we use the 20 nearest neighbors. We will open-source the complete baseline code implementation.

---

> > ### Comment · Reviewer_yq1V · 2025-08-04
> > **Answer to authors' rebuttal**
> >
> > I thank the authors for their rebuttal, which helped clarify several aspects of the paper. Nonetheless, I still have two major concerns that remain insufficiently addressed: (1) the significance of the task, and (2) the design and fairness of the experimental evaluation.
> >
> > ## Task significance
> > It appears there may have been a misunderstanding regarding my concern. My primary issue is with the limited predictive setup considered: forecasting only one day ahead using solely the previous day’s data. In real-world scenarios, models are typically expected to leverage longer historical windows and to support multi-step forecasting. This  would clearly frame the problem as a **spatio-temporal** forecasting task, and would justify the use of the considered baselines, which instead are not aligned with the current - mainly **spatial** - task. Also, the limited scope of the forecasting horizon (one day) could be better appreciated if considered at a finer temporal resolution, e.g., hourly, with a system able to forecast all the 24 hours in the horizon.
> >
> > Regarding the statement in the rebuttal that
> > > "station configurations in hourly settings often exhibit less dynamic variability, whereas the daily scale better captures the spatial-temporal patterns"
> >
> > could the authors please clarify what is meant by "spatial-temporal patterns" in this context? It is not clear why these patterns would be more evident in daily averages than in finer temporal resolutions, where temporal dynamics are presumably richer.
> >
> > ## Experimental evaluation
> > My second concern pertains to the experimental setup and the clarity of implementation details, some of which remain unclear even after the rebuttal. I would appreciate further clarification on the following points:
> > 1. The use of small batch sizes, with search limited to 1-2, might have a negative impact on the baselines. Did the authors try with larger batch sizes for these baselines?
> > 2. The additional results included in the rebuttal show better performance when predicting 2 steps ahead compared to 1 step, which is counter-intuitive. Could the authors provide an explanation for this behavior? Moreover, many baselines support direct multi-step prediction strategies, which generally outperform recursive approaches. Was this considered?
> > 3. My original Question 4, regarding how MIGN handles multiple time steps, remains under-addressed. The rebuttal does not make it clear how temporal dynamics are integrated into MIGN’s otherwise static architecture. Could the authors specify any modifications made to accommodate temporal inputs?
> > 4. The explanation of how positional or spatial embeddings are implemented in global-local baselines (such as T&S-AMP and HD-TTS) remains too high-level. Can the authors provide concrete implementation details and confirm whether any station-specific parameters or learned embeddings are used in these baselines?
> >
> > In conclusion, I find the proposed method promising and potentially impactful. However, without clearer justification of the task formulation and more detailed and transparent experimental design, it is difficult to properly assess the contribution or to ensure a fair comparison with existing work.

---

> ### Author Response · Authors · 2025-08-06
>
> We sincerely thank the reviewer for the reply! In the following, we provide the additional results and experimental details to address your concerns.
>
> > Task significance 1: forecasting only one day ahead using solely the previous day’s data.
>
>
> Thank you for raising this important point regarding task significance. To address your concern, we conducted additional experiments evaluating spatio-temporal forecasting performance.
>
> As shown in our input length study (Appendix), increasing the input steps from 3 to 4 yields only marginal performance gains. Therefore, we maintained the 3-step input configuration for both our model and baselines, while predicting 4 output steps (representing a medium-range 4-day forecast horizon). During inference, all 4 steps are generated in a single forward pass.
>
> Table 1(MSE computed over the entire 4-step sequence as a whole):
> |Model|MAX TEMP|MIN TEMP|DEWP|SLP|WDSP|MXSPD|
> |-------------|--------|--------|--------|--------|--------|--------|
> |Persistence|19.60|18.63|20.49|61.16|14.77|33.91|
> |STGCN| 17.52|16.08|17.48|42.86|10.31|24.58|
> |DyGrAE| 17.57|15.44|16.88|43.80|10.46|24.22|
> |TASAMP|17.43|16.34|18.21|43.54|10.87|24.39|
> |MIGN|14.45 |13.94|15.51|40.26|10.18|24.10|
>
> Table 2(MSE for step 1)：
> |Model|MAX TEMP|MIN TEMP|DEWP|SLP|WDSP|MXSPD|
> |-------------|--------|--------|--------|--------|--------|--------|
> |STGCN|9.10 | 9.38| 9.12 | 22.24|8.27  |  20.44|
> |DyGrAE| 9.91|9.14 |8.99  |23.47 |8.34  |20.33  |
> |TASAMP| 9.90|10.17 |9.17  |23.58 |8.30  |20.49  |
> |MIGN| 7.83|7.99 |7.69  | 19.67| 8.21 | 19.67 |
>
> Table 3(MSE for step 2)：
> |Model |MAX TEMP|MIN TEMP|DEWP|SLP|WDSP|MXSPD|
> |-------------|--------|--------|--------|--------|--------|--------|
> |STGCN|16.86 | 15.65| 17.14 |42.92 | 10.54 |  24.53|
> |DyGrAE|16.89 |15.32 |16.89 |45.29 |10.71  | 24.42 |
> |TASAMP|16.88 |15.58 |18.25  |43.72 |10.52  |24.55  |
> |MIGN|13.63 | 13.47| 14.99 | 40.81| 10.32 |24.11 |
>
>
> Table 4(MSE for step 3)：
> |Model |MAX TEMP|MIN TEMP|DEWP|SLP|WDSP|MXSPD|
> |-------------|--------|--------|--------|--------|--------|--------|
> |STGCN|20.08 | 18.36| 20.15 |51.08 | 11.10 |25.49  |
> |DyGrAE| 20.42|18.24 |20.19  |53.66 |11.23  |25.40  |
> |TASAMP| 20.14| 18.33|21.87  |51.90 |11.04  |25.50  |
> |MIGN|17.11 |16.37| 18.65 |49.68 | 10.88 | 25.31 |
>
>
> Table 5(MSE for step 4)：
> |Model |MAX TEMP|MIN TEMP|DEWP|SLP|WDSP|MXSPD|
> |-------------|--------|--------|--------|--------|--------|--------|
> |STGCN|22.05 |19.95 |  22.22|55.21 | 11.33 |26.01  |
> |DyGrAE| 22.49|19.78 |21.83  |58.28 |11.48  |25.98  |
> |TASAMP|22.17 |19.86 |23.84  |56.06 |11.32  |25.99  |
> |MIGN| 19.20|18.02 | 20.68 |53.93 | 11.12 | 25.85 |
>
> The results demonstrate that MIGN maintains superior performance compared to baseline methods in this multi-step forecasting setting.
>
> > Task significance 2: the limited scope of the forecasting horizon.
>
> We appreciate the reviewer's insightful comment regarding forecasting horizons. We would like to clarify that our framework can indeed support extended forecasting horizons, as demonstrated in our comparative analysis (see Experimental Evaluation 2). Below we address the relationship between hourly and daily forecasting:
>
> * **Empirical applications**: We acknowledge the critical importance of high-temporal-resolution (e.g., hourly) forecasts for applications requiring immediate response, such as severe weather warnings. Simultaneously, daily aggregated forecasts (e.g., max/min temperature, SLP) remain essential for strategic planning in agriculture, energy management, and climate adaptation.
> * **Task difficulty**: Daily extreme-value predictions (e.g., MAX/MIN temperature) represent physically integrated extremes over full diurnal cycles, not temporal averages. This constitutes a fundamentally different and often more challenging task than predicting instantaneous hourly values within smoother temporal sequences.
>
> We fully agree that multi-step hourly forecasting represents an important research direction. We have explicitly committed to extending MIGN for high-resolution temporal forecasting in future work and will strengthen this discussion in the revised manuscript.

---

> > ### Author Response · Authors · 2025-08-06
> >
> > > Task significance 3: spatial-temporal patterns.
> >
> > We sincerely appreciate this insightful question regarding temporal forecasting perspectives.
> >
> > Our framework focuses on predicting extreme-value targets (daily maximum/minimum temperature and maximum sustained wind speed) that are inherently defined at daily timescales. These physically significant extremes are not well-captured through hourly aggregation. Furthermore, daily forecasting aligns with medium-range prediction horizons (beyond 24 hours), while hourly forecasting typically serves short-term operational needs.
> >
> > We agree that extending to multi-step and longer-context forecasting represents valuable future work. To address this direction, we have conducted additional experiments examining:
> > - Autoregressive inference approaches
> > - Direct multi-step training paradigms
> >
> > These new results (presented in Experimental Evaluation 2 and Task Significance Analysis 1) demonstrate promising pathways for temporal extension. We will incorporate this discussion more prominently in the revised manuscript.
> >
> > From a spatial perspectivce, our selection of daily observations was driven by their superior station coverage compared to hourly data, which significantly enhances spatial pattern learning. According to NOAA's 2022 reports:
> > - Daily dataset: >10,000 stations/day (average)
> > - Hourly dataset: ~6,000 stations/hour (average)
> >
> > For building a robust global model, this increased station density is essential to:
> > 1. Capture diverse geographical features across underrepresented regions
> > 2. Improve characterization of spatial relationships
> > 3. Enhance generalization capabilities worldwide
> >
> > The expanded spatial coverage in daily data provides a more comprehensive foundation for learning the complex spatial dependencies central to our modeling approach.
> >
> > We thank the reviewer for this valuable observation and will clarify this spatial coverage advantage in the revised manuscript's Data Selection section.
> >
> > > Experimental evaluation 1: batch sizes.
> >
> > We appreciate the reviewer's insightful question regarding batch size selection.
> >
> > For HD-TTS method, we initially employed a batch size of 2 due to computational constraints (RTX 3090 GPU memory limitations).
> >
> > To rigorously examine batch size effects, we secured temporary access to an A800 GPU (80GB memory) and conducted additional experiments with HD-TTS across batch sizes [1, 2, 4, 8], maintaining identical hyperparameter configurations throughout:
> >
> > Table 6
> > |HD-TTS|MAX TEMP|MIN TEMP|DEWP|SLP|WDSP|MXSPD|
> > |-------------|--------|--------|--------|--------|--------|--------|
> > |Hidden State| 32| 64|128 | 64|32 |32 |
> > |Learning Rate|0.0007 | 0.0074| 0.0043| 0.0044|0.0012 |0.0012 |
> > |Batchsize 1| 10.25|9.63 | 9.52| 24.29|9.34 |20.91 |
> > |Batchsize 2| 10.31| 9.57| 9.81| 24.42|9.09 |20.97 |
> > |Batchsize 4|10.12 |9.49 |9.98 | 24.47| 9.13| 20.73|
> > |Batchsize 8| 10.34|9.52 | 9.75| 24.17| 9.56| 20.80|
> >
> > Results demonstrate stable performance across this range: While larger batches improve metrics like MAX TEMP/MIN TEMP, smaller batches benefit variables such as DEWP/WDSP. All configurations maintain competitive results within an acceptable performance threshold, confirming the model's robustness to batch size variations.

---

> ### Author Response · Authors · 2025-08-06
>
> > Experimental evaluation 2: results explanation.
>
> We sincerely appreciate the reviewer's careful attention to detail regarding to Autoregreeisve Table. We acknowledge that the original caption may have caused confusion about the evaluation metric. The caption “Table 7 (MSE): Trained with 1 input step and 1 output step; during inference, 4 output steps are generated autoregressively” refers to the MSE computed over the entire 4-step sequence as a whole, rather than the MSE for the first predicted step alone. The step-1 prediction results are already presented in the **main table**. For clarity, we also include the step-1 results again in the subsequent table for direct comparison.
>
> Table 7 (MSE) (Trained with 1 input step and 1 output step; during inference, 4 output steps are generated autoregressively (showing the MSE computed over the entire 4-step sequence as a whole))：
> |Model Variant|MAX TEMP|MIN TEMP|DEWP|SLP|WDSP|MXSPD|
> |-------------|--------|--------|--------|--------|--------|--------|
> |Persistence|19.60|18.63|20.49|61.16|14.77|33.91|
> |STGCN| 19.01|18.12|19.41|46.73|11.46|26.54|
> |DyGrAE| 18.77|19.15|19.38|46.24|11.83|26.87|
> |TASAMP| 20.59|37.27|20.39|45.92|12.71|28.30|
> |MIGN|15.79 |14.81|16.62|45.69|11.31|25.43
>
> Table 8(MSE for step 1)：
> |Model Variant|MAX TEMP|MIN TEMP|DEWP|SLP|WDSP|MXSPD|
> |-------------|--------|--------|--------|--------|--------|--------|
> |STGCN|9.74|9.44|9.25 |24.15| 8.60| 20.63|
> |DyGrAE|10.13  |9.49 |9.25  |24.09|8.77  |  20.78|
> |TASAMP|10.16  |12.90 |9.43  |24.38  |8.88  | 20.72 |
> |MIGN| 8.47 |8.01 |7.92  |20.09  |8.38  |19.73  |
>
> Table 9(MSE for step 2)：
> |Model Variant|MAX TEMP|MIN TEMP|DEWP|SLP|WDSP|MXSPD|
> |-------------|--------|--------|--------|--------|--------|--------|
> |STGCN|17.83 |17.26|18.75 |46.29 | 11.46| 26.21|
> |DyGrAE| 17.77| 17.98| 18.84| 46.07| 11.76| 26.52|
> |TASAMP|18.84 |31.44 |19.39 | 45.67|  12.80|28.42 |
> |MIGN| 14.51| 13.97| 15.70|42.57 |11.27 | 25.13|
>
> Table 10(MSE for step 3)：
> |Model Variant|MAX TEMP|MIN TEMP|DEWP|SLP|WDSP|MXSPD|
> |-------------|--------|--------|--------|--------|--------|--------|
> |STGCN|22.56 |21.43|23.31 |55.55 |12.47 | 28.69|
> |DyGrAE|22.29 | 22.74| 23.34|54.90 | 12.91| 28.99|
> |TASAMP|24.49 | 44.94|24.50 | 54.43|13.97 |30.45 |
> |MIGN|18.60 | 17.33| 19.90| 53.75| 12.39|27.55 |
>
> Table 11(MSE for step 4)：
> |Model Variant|MAX TEMP|MIN TEMP|DEWP|SLP|WDSP|MXSPD|
> |-------------|--------|--------|--------|--------|--------|--------|
> |STGCN|25.96|24.04|26.00 | 60.87| 13.34|30.64 |
> |DyGrAE| 25.23|25.95 |25.77 | 58.77|13.94 |31.05 |
> |TASAMP| 28.90|57.11 |27.84 | 59.27| 14.83|32.11 |
> |MIGN| 21.74|19.66 |22.63|58.12 | 13.30|29.37 |
>
>
> > Experimental evaluation 3: how temporal dynamics are integrated into MIGN’s otherwise static architecture?
>
> We sincerely appreciate this insightful technical question regarding MIGN's temporal integration mechanism. Our architecture processes temporal dynamics through the following sequence:
>
> **Independent Time-step Processing**:
>     For each input time step $t_{-N+1}, \ldots, t_{-1}, t_0$, we apply the Mesh Interpolation Encoder and perform Message Passing **independently**, generating static mesh graph features as defined in Equations (9)-(12).
>
> **Temporal Tensor Construction**:
>     With $K$ mesh nodes and $H$-dimensional features per time step, we obtain $N$ tensors of shape $[K, H]$. These are concatenated along the temporal dimension to form an input tensor of shape $[K, H, N]$.
>
> **Temporal Projection**:
>     A specialized multilayer perceptron (MLP) with parameters conditioned on $(N, M)$ maps the input tensor $[K, H, N]$ to decoder-ready features of shape $[K, H, M]$, effectively distributing temporal information across output steps.
>
> **Decoder Output Generation**:
> After obtaining the output mesh tensor $[K, H, M]$, we **independently** apply the Station Interpolation Decoder (Equation 13) to each $[K, H]$ feature slice corresponding to output time steps $t_1, t_2, \ldots, t_M$ to generate station-level predictions.
>
> This design maintains spatial processing efficiency while enabling flexible temporal modeling through the parameter-conditioned MLP bridge. We thank the reviewer for highlighting this architectural detail and will enhance the method part in the revised manuscript to clarify this temporal integration mechanism.

---

> ### Author Response · Authors · 2025-08-06
>
> > Experimental evaluation 4: how positional or spatial embeddings are implemented?
>
> We appreciate the reviewer's insightful question regarding our positional embedding approach.
>
> In baseline methods (T&S-AMP and HD-TTS), models learn station-specific parameters, yielding an embedding tensor of shape $[N, H]$ where $N$ is station count and $H$ is embedding dimension.
>
> Our approach differs fundamentally:
> 1.  **Geographic Encoding**:
>     We compute positional embeddings using geographic coordinates:
>     - Longitude $\lambda \in [-\pi, \pi]$
>     - Latitude $\theta \in [-\pi/2, \pi/2]$
>     Yielding $PE(\lambda, \theta) = (\lambda, \theta)$ → tensor shape $[N, 2]$
>
> 2.  **Shared Projection**:
>     Rather than station-specific embeddings, we apply a global linear transformation:
>     $\text{Linear}(2, H): [N, 2] \rightarrow [N, H]$
>
> This design:
> - Learns a universal embedding function
> - Enhances cross-station generalization
> - Maintains spatial awareness without station-specific parameters
>
> We thank the reviewer for this valuable inquiry and will clarify these architectural distinctions in the revised manuscript.

---

> > ### Author Response · Authors · 2025-08-07
> > **A kind reminder: the discussion period will end in two days**
> >
> > Dear Reviewers,
> >
> > We sincerely thank you for providing valuable feedback. We kindly remind you that the responses to your concerns have been posted. As the discussion period will end in two days, we would be grateful if you could allocate some time to review our responses.
> >
> > Thanks for all the reviewers' time again.
> >
> > Best regards,
> >
> > Authors

---

> > > ### Comment · Reviewer_yq1V · 2025-08-07
> > >
> > > Thank you for your thorough and thoughtful rebuttal. I appreciate the detailed clarifications and the effort to address the points I raised. A few remaining comments:
> > >
> > > * I find the multi-step prediction results more relevant than the one-step results presented in the paper. In particular, the direct approach (generating all 4 steps in a single forward pass) yields the best performance across models and is more aligned with standard practices in spatio-temporal forecasting. Please include these results **extended to all baselines** in the final version, ideally in the main body of the paper.
> > > * The discussion on the spatial and temporal resolution of the dataset (which motivates its choice) and the technical implementation details would benefit from the clarification provided in the rebuttal. Please consider expanding this in the final version.
> > >
> > > With the explanations and proposed revisions, I am satisfied that my concerns have been adequately addressed.

---

> > > > ### Author Response · Authors · 2025-08-07
> > > >
> > > > We sincerely thank you for providing valuable feedback.  In the final version, we will include the multi-step prediction results for all baselines in the main body of the paper and will expand the discussion on dataset characteristics and technical details in the Appendix.
> > > >
> > > > We are grateful for your acknowledgment that our rebuttals solve your concerns. If you are satisfied with our rebuttals, we kindly request your approval to consider an improved score to reflect the strengthened contributions of our work.
> > > >
> > > > Thank you again for your time and constructive input.

---

### Official Review · Reviewer_Fzb7 · 2025-07-02

**Clarity:** 3
**Significance:** 2
**Originality:** 2
**Rating:** 4
**Confidence:** 4

**Summary:**

This paper proposes a novel Mesh Interpolation Graph Network (MIGN) to address the issue of limited generalization ability in existing weather forecasting models, which is caused by finite and local areas for training and the neglect of the irregular and dynamic distribution of meteorological stations. MIGN effectively mitigates the spatial irregularity of meteorological station data through mesh interpolation and enhances global weather learning via parametric spherical harmonics location embedding, thereby endowing the model with robust generalization capabilities. Extensive experiments demonstrate MIGN's superior performance, spatial generalization ability, and capacity to generalize to previously unseen stations.

**Questions:**

1. As the authors mentioned in line 259, using the value from the previous day as the prediction for the next day has already outperformed most deep-learning models. Moreover, the benchmark methods chosen for the comparative experiments do not seem to include more novel approaches from the past two years, which somewhat undermines the persuasiveness of the experimental results.
2. Both in sections 3.2 and 4.2, the authors fail to provide a detailed rationale for choosing parametric spherical harmonics to learn geographical geometric information, instead of directly inputting coordinate information into the neural network to learn the location embedding. A more thorough explanation, supported by additional ablation studies, is needed to validate this choice.
3. Some of the equations in the paper are either redundant or not fully explained. For example, the significance of $n$ and $m$ in Equation 4 is not explained; Equations 9 and 11 contain redundant superscript $t$ with the same meaning, which could be further simplified. Moreover, Equation 9 does not explain how to determine the neighboring meteorological station nodes for the regular grid, and this needs to be clarified.
4. When presenting the model performance, the authors should indicate the specific percentage of improvement rather than simply listing the corresponding metric results.

**Ethical Concerns:**

["NO or VERY MINOR ethics concerns only"]

**Final Justification:**

The rebuttal has partially addressed my concerns. However, the baseline methods included are still not representative of the most widely adopted state-of-the-art approaches in spatiotemporal forecasting. Additionally, several of the baselines are designed for classification rather than forecasting, which undermines the relevance and strength of the experimental comparisons. The proposed method also lacks originality, as it primarily applies existing techniques without significant novel contributions. Given these considerations, I will maintain my original score.

**Limitations:**

Yes. The authors discussed the limitations of their work in the conclusion section and elaborated on the Broader Impacts in the Appendix A.1.

**Paper Formatting Concerns:**

There are no significant formatting issues in this article.

**Quality:**

3

**Strengths And Weaknesses:**

Strengths:
1. The spatial correlations among irregularly and dynamically distributed meteorological stations are modeled through message passing between meteorological station nodes and regular grid nodes.
2. By employing parametric spherical harmonics location embedding to encode the coordinates of meteorological stations, the model is empowered to generalize to unobserved areas.
3. Through extensive comparative and ablation experiments, this paper comprehensively validates the superior performance and robust generalization capability of MIGN, especially in sparse regions and newly added meteorological stations.

Weaknesses:
1. As the authors mentioned in line 259, using the value from the previous day as the prediction for the next day has already outperformed most deep-learning models. Moreover, the benchmark methods chosen for the comparative experiments do not seem to include more novel approaches from the past two years, which somewhat undermines the persuasiveness of the experimental results.
2. Both in sections 3.2 and 4.2, the authors fail to provide a detailed rationale for choosing parametric spherical harmonics to learn geographical geometric information, instead of directly inputting coordinate information into the neural network to learn the location embedding. A more thorough explanation, supported by additional ablation studies, is needed to validate this choice.
3. Some of the equations in the paper are either redundant or not fully explained. For example, the significance of  $n$ and $m$ in Equation 4 is not explained; Equations 9 and 11 contain redundant superscript $t$ with the same meaning, which could be further simplified. Moreover, Equation 9 does not explain how to determine the neighboring meteorological station nodes for the regular grid, and this needs to be clarified.

---

> ### Author Rebuttal · Authors · 2025-07-31
>
> We appreciate the listing of our paper's strengths and have addressed all questions below.
>
> >Weaknesses 1 & Questions 1: Latest spatial-temporal baseline
>
> Thank you for the suggestion. We have incorporated additional recent spatiotemporal forecasting baselines, including DualCast [1] and ReDyNet [2]. MIGN consistently outperforms these methods as well, further demonstrating its effectiveness.
>
> MSE
> |Model Variant|MAX TEMP|MIN TEMP|DEWP|SLP|WDSP|MXSPD|
> |-------------|--------|--------|--------|--------|--------|--------|
> |DualCast|9.92|9.68|9.53|23.98|8.54|19.92
> |ReDyNet|10.65|9.42|9.36|24.34|8.77|21.03
> |MIGN|8.55 |8.05|7.95|20.90|8.34|19.82|
>
> MAE
> |Model Variant|MAX TEMP|MIN TEMP|DEWP|SLP|WDSP|MXSPD|
> |-------------|--------|--------|--------|--------|--------|--------|
> |DualCast|2.24|2.17|2.15|3.41|2.02|3.26|
> |ReDyNet|2.37|2.12|2.11|3.41|2.06|3.31|
> |MIGN|2.10 |2.00|1.99|3.14|1.98|3.20|
>
> [1] DualCast: A Model to Disentangle Aperiodic Events from Traffic Series. IJCAI 2025.
> [2] Responsive Dynamic Graph Disentanglement for Metro Flow Forecasting. AAAI 2025.
>
> >Weaknesses 2 & Questions 2: Ablation study of location encoding.
>
> Thanks for the suggestion. We further compare our SH embedding method with three commonly used coordinate-based location embedding approaches, including Direct Coordinate, WRAP, and Cartesian 3D. The details are as follows:
> Consider longitude $\lambda \in [-\pi,\pi]$ and latitude $\theta \in [-\pi/2,\pi/2]$
> Direct: $PE(\lambda, \theta) = (\lambda, \theta)$
> WRAP: $PE(\lambda, \theta) = [cos\lambda,sin\lambda, cos\theta, sin\theta]$
> CARTESIAN3D:  $PE(\lambda, \theta) =[cos\theta cos\lambda,cos\theta sin\lambda,sin\theta ]$
>
> We apply the three location embedding methods to both the encoder and decoder of our model. Our proposed SH embedding strategy outperforms the other three baseline methods, demonstrating its effectiveness.
>
>
>
> MSE
> |Model Variant|MAX TEMP|MIN TEMP|DEWP|SLP|WDSP|MXSPD|
> |-------------|--------|--------|--------|--------|--------|--------|
> |W/O|9.04|8.71|8.71|23.01|8.76|20.63|
> |Direct|8.88|8.57|8.35|21.89|8.64|19.85|
> |WRAP|8.70|8.32|8.23|21.14|8.52|19.83|
> |CARTESIAN3D|8.67|8.34| 8.19|21.21|8.48|19.86|
> |SH Embedding|8.47|8.01|7.92|20.09|8.38|19.73|
>
>
>
> >Weaknesses 3 & Questions 3: Some of the equations in the paper are either redundant or not fully explained.
>
> Thanks for the suggestion. We will revise the equations in the revised version of the paper.
>
> > Questions 4: Indicating the specific percentage of improvement.
>
> Thank you for the suggestion. We have included the specific percentage improvements shown in the following table in the revised version of the paper.
>
>
> Main table
> |MSE|MAX TEMP|MIN TEMP|DEWP|SLP|WDSP|MXSPD|
> |-------------|--------|--------|--------|--------|--------|--------|
> |Overall Improvement|13%| 15%| 14%| 16%| 2%|1% |
>
>
>
> Ablation study
> |MSE|MAX TEMP|MIN TEMP|DEWP|SLP|WDSP|MXSPD|
> |-------------|--------|--------|--------|--------|--------|--------|
> |Mesh Improvement|10%|11% |12% |16%  |4% |4% |
> |SH Embedding Improvement|6%|8.03%|9%|12%|4%|4%

---

> > ### Comment · Reviewer_Fzb7 · 2025-08-08
> > **Reply to the rebuttal**
> >
> > The rebuttal has partially addressed my concerns. However, the baseline methods included are still not representative of the most widely adopted state-of-the-art approaches in spatiotemporal forecasting. Additionally, several of the baselines are designed for classification rather than forecasting, which undermines the relevance and strength of the experimental comparisons. The proposed method also lacks originality, as it primarily applies existing techniques without significant novel contributions. Given these considerations, I will maintain my original score.

---

### Official Review · Reviewer_fq8d · 2025-07-03

**Clarity:** 3
**Significance:** 3
**Originality:** 2
**Rating:** 4
**Confidence:** 4

**Summary:**

This paper introduces the Mesh Interpolation Graph Network (MIGN), a new framework for global weather forecasting using irregular and dynamically varying weather station data. The key contributions are (1) a mesh-based encoder-decoder architecture that interpolates irregular station data onto a regular mesh and performs message passing there, and (2) the use of parametric spherical harmonics to embed station locations, enabling better spatial generalization. The model is evaluated on the NOAA GSOD dataset and shows strong performance against 13 baselines. It also demonstrates superior generalization to unseen stations and robustness in data-scarce regions.

**Questions:**

1. How does the model perform when provided with longer historical input sequences (e.g., 3–5 days)? Is the current architecture capable of forecasting multiple steps into the future? Additionally, since the paper emphasizes the challenge of dynamically changing station distributions, could the authors elaborate on how the model handles situations where the spatial distribution shifts during multi-step inference?
2. How does the model perform in regions with no stations, such as oceans (as you mentioned in limitations)? Would MIGN still provide meaningful forecasts in these areas, and how would its performance compare to grid-based models like GraphCast? More broadly, can station-based models like MIGN be better compared to grid-based models? For example, given historical data or overlapping regions, is it feasible to compare their forecasts either locally (e.g., at specific locations) or globally (e.g., averaged over the domain)?

**Ethical Concerns:**

["NO or VERY MINOR ethics concerns only"]

**Final Justification:**

The rebuttal has addressed most of my concerns, and I find the proposed method well justified within this problem setup. However, I remain uncertain about its practical value, given its weak performance in non-stationary areas.

**Limitations:**

No, please see questions.

**Quality:**

3

**Strengths And Weaknesses:**

## Strengths
1. The paper addresses a meaningful problem in weather forecasting — how to model irregular and dynamic sensor data at global scale.
2. The encoder–processor–decoder design is intuitive and effective for this task.
3. The empirical section is extensive. The model consistently outperforms all baselines across a wide range of metrics. The generalization experiments to unseen stations and performance breakdowns by region are especially interesting.


## Weaknesses
1. The method forecasts one step ahead, and the architecture appears to focus almost entirely on spatial modeling. It’s not clear how well this would extend to multi-step forecasting or longer temporal horizons, which are important for operational forecasting use cases.
2. While spherical harmonics are theoretically sound, the paper does not provide much intuition or visualization of what the learned SH embeddings capture.
3. It’s not fully clear how the model scales with increasing station density or mesh resolution. A brief introduction on training/inference cost would help contextualize the method’s practicality.

---

> ### Author Rebuttal · Authors · 2025-07-31
>
> We sincerely thank the reviewer for recognizing the studied problem, method effectiveness, and extensive experiments. We have addressed all questions below.
>
> >Weaknesses 1: Extend to Multi-step forecasting.
>
>
> Thank you for raising the issue. We agree that multi-step autoregressive assessment would provide stronger evidence of the model's predictive capabilities. Thus, we additionally evaluate the iterative performance of single-step models in generating 4-day predictions. The results are presented in the following table.
>
>
> Table 1(MSE):Trained with 1 input step and 1 output step; during inference, 4 output steps are generated autoregressively：
> |Model Variant|MAX TEMP|MIN TEMP|DEWP|SLP|WDSP|MXSPD|
> |-------------|--------|--------|--------|--------|--------|--------|
> |Persistence|19.60|18.63|20.49|61.16|14.77|33.91|
> |STGCN| 19.01|18.12|19.41|46.73|11.46|26.54|
> |DyGrAE| 18.77|19.15|19.38|46.24|11.83|26.87|
> |TASAMP| 20.59|37.27|20.39|45.92|12.71|28.30|
> |MIGN|15.79 |14.81|16.62|45.69|11.31|25.43
>
> Table 2(MSE) for step 2：
> |Model Variant|MAX TEMP|MIN TEMP|DEWP|SLP|WDSP|MXSPD|
> |-------------|--------|--------|--------|--------|--------|--------|
> |STGCN|17.83 |17.26|18.75 |46.29 | 11.46| 26.21|
> |DyGrAE| 17.77| 17.98| 18.84| 46.07| 11.76| 26.52|
> |TASAMP|18.84 |31.44 |19.39 | 45.67|  12.80|28.42 |
> |MIGN| 14.51| 13.97| 15.70|42.57 |11.27 | 25.13|
>
> Table 3(MSE) for step 3：
> |Model Variant|MAX TEMP|MIN TEMP|DEWP|SLP|WDSP|MXSPD|
> |-------------|--------|--------|--------|--------|--------|--------|
> |STGCN|22.56 |21.43|23.31 |55.55 |12.47 | 28.69|
> |DyGrAE|22.29 | 22.74| 23.34|54.90 | 12.91| 28.99|
> |TASAMP|24.49 | 44.94|24.50 | 54.43|13.97 |30.45 |
> |MIGN|18.60 | 17.33| 19.90| 53.75| 12.39|27.55 |
>
> Table 4(MSE) for step 4：
> |Model Variant|MAX TEMP|MIN TEMP|DEWP|SLP|WDSP|MXSPD|
> |-------------|--------|--------|--------|--------|--------|--------|
> |STGCN|25.96|24.04|26.00 | 60.87| 13.34|30.64 |
> |DyGrAE| 25.23|25.95 |25.77 | 58.77|13.94 |31.05 |
> |TASAMP| 28.90|57.11 |27.84 | 59.27| 14.83|32.11 |
> |MIGN| 21.74|19.66 |22.63|58.12 | 13.30|29.37 |
>
> From the results, we can find that:
>
> * MIGN achieves the lowest MSE on the total errors, demonstrating its effectiveness.
> * MIGN demonstrates superior performance compared to the baselines when evaluated under conditions of error accumulation.
>
> These results further verify the effectiveness of MIGN. Thank you for motivating us to conduct the experiments.
>
>
>
>
> > Weekness 2: Visualization of the learned SH embeddings.
>
> Thanks for the comment. Due to limitations of the rebuttal format, we are unable to include visualizations here. However, in the revised version of the paper, we will include comprehensive visualizations of each SH embedding dimension by projecting station locations onto the sphere and coloring them according to the corresponding embedding values.
>
> > Weekness 3:  Training/inference cost.
>
> Thank you for the suggestion. The computational cost analysis is provided in Appendix A.7.  Mesh interpolation is achieved by constructing a nearest neighbors graph between feature nodes and Healpix nodes. Suppose there are $M$ feature nodes and $N$ Heslpix nodes. Using a brute-force approach, the computational complexity is $O(MN)$. For baselines, the computational complexity of nearest neighbor connections among the feature nodes is $O(M^2)$. We found that the optimal number of Healpix nodes $N$ is smaller than the number of feature nodes $M$, meaning that $N<M$, thus $O(MN)<O(M^2)$. Therefore, this interpolation step is more efficient than directly constructing nearest-neighbor graphs among all station nodes, as typically done in standard GNN baselines. Moreover, our model adopts a simple message passing mechanism; thus, from a theoretical perspective, its overall complexity is comparable to that of commonly used spatio-temporal GNNs.
>
> To further demonstrate the training and inference efficiency of our model, we compare the training and inference times per step of all models on an NVIDIA RTX 3090 GPU, as shown in the following table. We observe that the training and inference time of MIGN is comparable to that of STGCN and MPNNLSTM, demonstrating its efficiency and practical effectiveness.
>
> |Model|STGCN|TGCN|DyGrAE |MPNNLSTM |GPS |HD-TTS|MIGN|
> |-------------|--------|--------|--------|--------|--------|--------|--------|
> |Training time per step(s)|0.013|0.014|0.016|0.012|0.048|3.25|0.013|
> |Inference time per step(s)|0.004|0.010|0.012|0.011|0.019|3.03|0.006|
>
>
>
> > Questions 1: Model performance w.r.t.  longer historical input sequences.
>
> Thanks for your comment. The further input step analysis are shown in Appendix A.6. MIGN consistently outperforms all other models across input steps ranging from 1 to 4. We find that increasing the input step from 3 to 4 yields only marginal gains for most models.
>
> > Questions 1: Is the current architecture capable of forecasting multiple steps into the future?
>
> Yes, please see Weeknesses 1.
>
> > Questions 1: How the model handles situations where the spatial distribution shifts during multi-step inference?
>
> We appreciate the reviewer's valuable comment regarding our inference process. In our multi-step inference framework, we employ the following procedure: (1) dynamically varying station observations are first projected onto static mesh nodes through mesh-based interpolation; (2) spatial message passing is then performed across the mesh nodes; and (3) finally, the decoder transforms the mesh node embeddings back to the target station distribution for prediction.
>
>
> > Questions 2: Model performance w.r.t. regions with no stations.
>
> We sincerely appreciate the reviewer's insightful question regarding model performance in unobserved regions. To systematically evaluate this capability, we conducted rigorous regional generalization experiments (detailed in Appendix A.6), where models were trained exclusively on data from observed regions and tested on completely unobserved areas. Baseline approaches including TGCN, DyGrAE, and MPNN-LSTM exhibit limited forecasting capability in unobserved regions, with minimum temperature prediction MAE values reaching 7.61°C, 5.32°C, and 13.60°C respectively - performance levels that may not meet practical application requirements. In contrast, MIGN consistently outperforms the baseline models, demonstrating its stronger ability to generalize to areas without direct observations.
>
> > Questions 2: How would its performance compare to grid-based models like GraphCast? More broadly, can station-based models like MIGN be better compared to grid-based models?
>
> In weather forecasting, approaches are generally categorized into two paradigms based on their input data: (1) Gridded reanalysis data: This is the foundation for models like Pangu-Weather and GraphCast, which operate on high-resolution global grids analysis data with 6-hour forecast intervals. (2) Discrete station observations: This task, which our work addresses, focuses on modeling sparse observational data using spatiotemporal techniques.
>
> While a technical comparison between GraphCast and MIGN could be attempted, such a comparison is not practically meaningful due to fundamental differences in their input data and application settings. GraphCast relies on gridded reanalysis data derived through data assimilation, whereas MIGN is designed to work with raw station-based observational data. Additionally, the two models differ in temporal resolution—GraphCast performs 6-hour forecasts, while MIGN operates at a daily timescale. Our work specifically addresses the station-based forecasting scenario, which involves unique challenges such as sparse spatial coverage, irregular station distribution, and observational noise. These characteristics distinguish it from grid-based forecasting and highlight the need for a tailored modeling approach.

---

> > ### Comment · Reviewer_fq8d · 2025-08-05
> >
> > Thank you for the additional results and clarification. I appreciate the new experiments on multi-step inference - this significantly strengthens the submission, and most of my concerns have been addressed.
> >
> > I only have one remaining concern regarding the comparison between methods based on gridded reanalysis data and those using discrete station observations. While I understand that station-based methods present unique challenges, gridded reanalysis data offers broader spatial coverage—including regions without stations—and finer temporal resolution (e.g., 6-hourly vs. daily). It is unclear for me why station-based methods are still necessary or preferable in certain contexts.
> >
> > In my view, a direct comparison between the two approaches is essential to justify the necessity of a station-based method.
> >
> > Therefore, I will maintain my original score.

---

> > > ### Author Response · Authors · 2025-08-06
> > > **Comparison between station-based method and gridded reanalysis approach**
> > >
> > > Thank you for your valuable feedback on station-based versus gridded reanalysis approaches. We appreciate your recognition of reanalysis data's strengths in spatial coverage and temporal resolution for data-sparse regions. We agree both methodologies offer complementary value and will enhance our manuscript's discussion accordingly. Our perspective on station-based methods' importance is grounded in three key aspects:
> > >
> > > 1.  **Observation Dependence**:
> > >     Reanalysis fundamentally relies on assimilating station observations—without high-quality station data, reanalysis outputs would lack reliability.
> > > 2.  **Complementary Strengths**:
> > >     Station-based methods excel at capturing fine-scale spatial variability critical for extreme weather prediction and urban forecasting, while offering practical advantages for real-time deployment in regions with limited reanalysis access.
> > > 3.  **Computational Efficiency**:
> > >     Station-based training is significantly less intensive than processing high-dimensional (4D) gridded data, enabling strong performance with lower hardware requirements—enhancing accessibility in resource-constrained environments.
> > >
> > > **Empirical Performance Comparison**:
> > > We quantitatively compared methods by bilinearly interpolating Pangu's 2022 gridded forecasts (WeatherBench2) to station locations. Our station-based model (MIGN) was trained on 2017-2020 data, validated on 2021, and evaluated against Pangu on 2022 data:
> > >
> > > | Model | MAX TEMP | MIN TEMP | WDSP |
> > > | :---- | :------- | :------- | :--- |
> > > | Pangu | 10.84    | 9.95     | 9.76 |
> > > | MIGN  | **8.71** | **9.02** | **8.60** |
> > >
> > > These results demonstrate MIGN's consistent advantage, highlighting the benefit of modeling directly on fine-grained station observations to capture local variability and extremes that may be smoothed in gridded outputs.

---

> > > > ### Comment · Reviewer_fq8d · 2025-08-07
> > > >
> > > > Thank you for the detailed response and clarification. The comparison results are helpful. I wonder—have you also considered interpolating the station-based forecasts to evaluate performance in locations without direct station coverage? It would be more interesting for me to compare how both approaches generalize spatially, especially in regions lacking station observational data.

---

> > > > > ### Author Response · Authors · 2025-08-07
> > > > >
> > > > > We sincerely appreciate this insightful query regarding model performance in unobserved regions. To address this: We randomly selected 10% of observation locations as ground truth points without direct station measurements. The remaining 90% of stations trained our station-based model. Both Pangu and MIGN 2022 predictions were bilinearly interpolated to these unmonitored locations.
> > > > >
> > > > > **Performance Comparison**:
> > > > >
> > > > > | Model                  | MAX TEMP | MIN TEMP | WDSP  |
> > > > > | :--------------------- | :------- | :------- | :---- |
> > > > > | Pangu (0.25° resolution) | 11.45    | 10.45    | 9.98  |
> > > > > | Pangu (1.00° resolution) | 14.12    | 13.25    | 12.87 |
> > > > > | MIGN                   | 13.84| 12.87| 13.14 |
> > > > >
> > > > > MIGN outperforms the 1.00° resolution Pangu on MAX TEMP and MIN TEMP and achieves competitive WDSP performance.
> > > > >
> > > > > While MIGN does not surpass the higher-resolution Pangu model (0.25° grid spacing), this is primarily due to Pangu’s substantial data advantages. Specifically, Pangu is trained on over 1 million global reanalysis points per snapshot, leveraging multiple 3D meteorological variables—such as geopotential height, temperature, humidity, and wind—across various pressure levels. This rich and high-resolution input enables strong predictive performance, but requires more than 200 TB of data.
> > > > >
> > > > > In contrast, MIGN achieves competitive results with significantly fewer input variables, utilizing only ~10,000 data points per day and a total data volume of around 10 GB. This advantage position MIGN as a lightweight and accessible alternative, especially under data-scarce or resource-constrained conditions.

---

> > > > > > ### Comment · Reviewer_fq8d · 2025-08-08
> > > > > >
> > > > > > Thank you for providing the additional results. I agree that it is acceptable for the proposed method to perform worse than Pangu under such setup. I highly recommend discussing this limitation and incorporating these comparison results into the paper if it is accepted.

---

> > > > > > > ### Author Response · Authors · 2025-08-08
> > > > > > >
> > > > > > > We sincerely thank you for providing valuable feedback. In the final version, we will discuss this limitation and incorporate the comparison results in the revised paper.
> > > > > > >
> > > > > > > We are grateful for your acknowledgment that our rebuttals solve your concerns. If you are satisfied with our rebuttals, we kindly request your approval to consider an improved score to reflect the strengthened contributions of our work.
> > > > > > >
> > > > > > > Thank you again for your time and constructive input.

---

### Official Review · Reviewer_AvaR · 2025-07-03

**Clarity:** 3
**Significance:** 3
**Originality:** 3
**Rating:** 4
**Confidence:** 4

**Summary:**

This paper presents a novel framework for global weather forecasting that addresses two fundamental challenges: spatially irregular weather station distributions and temporal variability in station availability. The approach combines mesh interpolation for mapping irregular observations onto a regular HEALPix grid with spherical harmonics location embedding for enhanced spatial generalization. Evaluated on the NOAA GSOD dataset against 13 baseline methods, MIGN demonstrates superior performance across multiple meteorological variables.

**Questions:**

See weaknesses.

**Ethical Concerns:**

["NO or VERY MINOR ethics concerns only"]

**Final Justification:**

The rebuttal has partially addressed my concerns. Hence, I keep my rating unchanged, which is borderline accept.

**Limitations:**

The authors have adequately addressed the limitations of their work.

**Quality:**

3

**Strengths And Weaknesses:**

Strengths:
- The paper is well-organized and clearly written, with a logical flow from problem formulation to experimental validation.
- The experimental evaluation and ablation analysis are comprehensive. The model's superior performance in data-scarce regions (e.g., Africa, South America) is particularly noteworthy for operational forecasting applications.

Weaknesses:
- The evaluation is limited to single-step forecasting, while multi-step autoregressive assessment would provide stronger evidence of the model's predictive capabilities, especially regarding error accumulation over extended horizons.
- The Global Generalization Analysis presents counterintuitive results that require clarification. Despite training on only half of the randomly sampled stations, several baseline models (HD-TTS, T&S-IMP, STAR) demonstrate improved performance compared to full dataset experiments (Table 3 vs. Table 1). For instance, HD-TTS shows reduced MSE for MAX TEMP (10.20 → 9.81), along with improvements in DEWP and WDSP metrics.

---

> ### Author Rebuttal · Authors · 2025-07-31
>
> Thank you for recognizing the novelty, comprehensive experiments, and writing of our paper. We have addressed all questions below.
>
> >Weaknesses 1: Multi-step autoregressive assessment.
>
> Thank you for raising the issue. We agree that multi-step autoregressive assessment would provide stronger evidence of the model's predictive capabilities. Thus, we additionally evaluate the iterative performance of single-step models in generating 4-day predictions. The results are presented in the following table.
>
> Table 1(MSE):Trained with 1 input step and 1 output step; during inference, 4 output steps are generated autoregressively：
> |Model Variant|MAX TEMP|MIN TEMP|DEWP|SLP|WDSP|MXSPD|
> |-------------|--------|--------|--------|--------|--------|--------|
> |Persistence|19.60|18.63|20.49|61.16|14.77|33.91|
> |STGCN| 19.01|18.12|19.41|46.73|11.46|26.54|
> |DyGrAE| 18.77|19.15|19.38|46.24|11.83|26.87|
> |TASAMP| 20.59|37.27|20.39|45.92|12.71|28.30|
> |MIGN|15.79 |14.81|16.62|45.69|11.31|25.43
>
> Table 2(MSE) for step 2：
> |Model Variant|MAX TEMP|MIN TEMP|DEWP|SLP|WDSP|MXSPD|
> |-------------|--------|--------|--------|--------|--------|--------|
> |STGCN|17.83 |17.26|18.75 |46.29 | 11.46| 26.21|
> |DyGrAE| 17.77| 17.98| 18.84| 46.07| 11.76| 26.52|
> |TASAMP|18.84 |31.44 |19.39 | 45.67|  12.80|28.42 |
> |MIGN| 14.51| 13.97| 15.70|42.57 |11.27 | 25.13|
>
> Table 3(MSE) for step 3：
> |Model Variant|MAX TEMP|MIN TEMP|DEWP|SLP|WDSP|MXSPD|
> |-------------|--------|--------|--------|--------|--------|--------|
> |STGCN|22.56 |21.43|23.31 |55.55 |12.47 | 28.69|
> |DyGrAE|22.29 | 22.74| 23.34|54.90 | 12.91| 28.99|
> |TASAMP|24.49 | 44.94|24.50 | 54.43|13.97 |30.45 |
> |MIGN|18.60 | 17.33| 19.90| 53.75| 12.39|27.55 |
>
> Table 4(MSE) for step 4：
> |Model Variant|MAX TEMP|MIN TEMP|DEWP|SLP|WDSP|MXSPD|
> |-------------|--------|--------|--------|--------|--------|--------|
> |STGCN|25.96|24.04|26.00 | 60.87| 13.34|30.64 |
> |DyGrAE| 25.23|25.95 |25.77 | 58.77|13.94 |31.05 |
> |TASAMP| 28.90|57.11 |27.84 | 59.27| 14.83|32.11 |
> |MIGN| 21.74|19.66 |22.63|58.12 | 13.30|29.37 |
>
> From the results, we can find that:
>
> * MIGN achieves the lowest MSE on the total errors, demonstrating its effectiveness.
> * MIGN demonstrates superior performance compared to the baselines when evaluated under conditions of error accumulation.
>
> These results further verify the effectiveness of MIGN. Thank you for motivating us to conduct the experiments.
>
> >Weaknesses 2: Clarification for Global Generalization Analysis.
>
> Thanks for raising these observations. Global generalization experiments evaluate the generalization ability to unseen stations. Several models perform worse when trained on the full dataset compared to the global generalization setting. This suggests that training on the full dataset may lead to overfitting to specific station patterns, resulting in poor generalization even to future observations at those same stations. For example, HD-TTS constructs downsampled graphs based on graph pooling. In dense regions (e.g., North America/EU), the full training set creates highly connected subgraphs. Models may over-rely on redundant local messages while underutilizing global patterns. Randomly subsampling stations thins these dense subgraphs, inadvertently encouraging longer-range dependency learning.

---

### Note · Authors · 2025-08-12

Dear Reviewers,

We sincerely appreciate your valuable feedback and dedicated time during the discussion stage. We're grateful for your recognition (`fq8d` and `yq1v`) of addressing a significant challenge in weather forecasting, while multiple reviewers (` AvaR`, `fq8d`, `Fzb7`, and `yq1v`) highlighted the comprehensive scope of our experimental evaluation. We're particularly grateful for commendation ( `AvaR`) of the manuscript's clarity and writing quality, and for recognition (`yq1v`) of our well-designed figures that effectively communicate key findings.


According to the questions and concerns raised by reviewers, we will add following point in the revised paper:

- Additional autoregressive multistep inference results (`AvaR`, `fq8d`, `yq1v`).
- Additional direct multistep training results (`yq1v`).
- Additional visualizations of the learned SH embeddings (`fq8d`).
- Additional results for Pangu-Weather and a discussion on the limitations of direct observation-based prediction models compared to gridded models (`fq8d`).
- Additional results from the latest spatio-temporal baselines (`Fzb7`).
- Additional ablation study on location embeddings (`Fzb7`).
- Additional discussion on the rationale for dataset selection and technical implementation details (`yq1v`).
- Additional explanations and revisions of the equations (`Fzb7`, `yq1v`).
- Additional description of the temporal integration mechanism in our method (`yq1v`).
- Additional results showing specific percentage improvements (`Fzb7`).


We sincerely appreciate your valuable insights, which have significantly contributed to improving the quality of our manuscript.

Best regards,

Authors.

---

### Decision · Program_Chairs · 2025-09-17

**Decision:**

Accept (poster)

**Comment:**

The paper proposes a method for global weather forecasting based on graph neural networks.

The strengths of the paper include a well-justified model based on message passing among both regular and irregular nodes, supporting direct observational data from irregularly located stations, an additional spherical harmonics coupling that enables improved generalization to locations with sparse observations, extensive empirical evaluation, state-of-the-art performance, and generally speaking good presentation.

In the reviews, the reviewers unanimously considered only 1-day ahead forecasting a weakness.

In the rebuttal, the authors provided extended results covering up to 4-day ahead forecasting task, where the method was still found to achieve state-of-the-art performance.

All reviewers found the rebuttal satisfactory and no remaining major issues remained. On the other hand, no reviewer was particularly enthusiastic about the work and all suggest only "borderline accept" suggesting that there is still room for improvement.

Lastly, it must be emphasized that formatting rules must be followed exactly without any tricks with negative vspace, overly small font-sizes, etc. Violating the rules risks desk-rejecting the paper. If the page limit becomes an issue, content that isn't essential to the main contribution can be included in appendices.